# Reconstruction of human dispersal during Aurignacian on pan-European scale

Yaping Shao [1] ✉, Christian Wegener [1], Konstantin Klein[1], Isabell Schmidt [2] & Gerd-Christian Weniger[2]

The Aurignacian is the first techno-complex related with certainty to Anatomically Modern Humans in Europe. Studies show that they appeared around 43-42 kyr cal BP and dispersed rapidly in Europe during the Upper Palaeolithic. However, human dispersal is a highly convoluted process which is until today not well understood. Here, we provide a reconstruction of the human dispersal during the Aurignacian on the pan-European scale using a human dispersal model, the Our Way Model, which combines archaeological with paleoclimate data and uses the human existence potential as a unifying driver of human population dynamics. Based on the reconstruction, we identify the different stages of the human dispersal and analyse how human demographic processes are influenced by climate change and topography. A chronology of the Aurignacian human groups in Europe is provided, which is verified for locations where archaeological dating records are available. Insights into highly debated hypotheses, such as human dispersal routes, are provided.

In Europe, the Aurignacian (AUR) is the first techno-complex that can be related to Anatomically Modern Humans (AMHs) that successfully spread to western and southwestern Europe[1]. Before the AUR, considering the palaeogenetic data from Oase 1[2], Bacho Kiro[3] and Zlatý kůň[4], at least two dispersal impulses reached eastern Europe but did not lead to a significant settlement for reasons still unknown[5]. Only from Bacho Kiro was a small lithic inventory recovered, attributed to the Initial Upper Palaeolithic (IUP). Studies on the Châtelperronian (CHÂT) in northern Spain also indicate a collapse of populations and subsequent recolonisation after the Middle Palaeolithic by the Neanderthal (NEA) groups of the CHÂT[6]. The situation of the IUP in the Rhone Valley with the Neronian is even more complex. The data from the Grotte Mandrin attributed to AMH[7] are contrasted by the findings from the Abri du Maras[8], which postulate an attribution of the Neronian to NEAs. In Italy, the IUP complex of the Uluzzian has been connected to AHM due to human remains at one site only[9]. The term IUP thus vaguely describes a heterogeneous group of lithic inventories attributed to both AMHs and NEAs, depending on region and research team. The technological definition and duration of the IUP are also highly controversial. In contrast, the definition of the AUR is more homogeneous, although it is not free of dispute, providing a solid starting point for a reconstruction of the AMH dispersal in Europe. We therefore deliberately limit our study here to the AUR, but will discuss in the Supplementary Discussion the possible influences of the IUP populations on the dispersal of humans of the AUR in Europe.

Conventionally, the AUR is divided into the Proto, Early, Evolved and Late Aurignacian, but the technological and chronological differences between the first two periods are difficult to distinguish[10,11]. We follow Schmidt and Zimmermann[12] and divide the techno-complex into Phase 1 (AUR-P1, 43–38 kyr cal BP; kyr cal BP hereafter ka), comprising the Proto and Early Aurignacian periods, and Phase 2 (AUR-P2, 38–32 ka), comprising the Evolved and Late Aurignacian periods.

It has been suggested that human groups with AUR technology first appeared in Europe around 43–42 ka[13] or earlier and then rapidly expanded during the Upper Palaeolithic[14]. The origin of the AUR is still being debated and a wide geographical route stretching from the Levant to the Black Sea or even further east has been considered to be possible[15]. Mellars[16] hypothesised that the AUR expansion started somewhere in Turkey and progressed in a major event during the AUR-P1, most likely during interstadial times. Based on the distribution and chronology of the archaeological sites in the Balkan and Upper Danube areas, he proposed approximate Danube and Mediterranean-Coast

[1]Institute for Geophysics and Meteorology, University of Cologne, Cologne, Germany. [2]Institute of Prehistory, University of Cologne, Cologne, Germany. ✉e-mail: yshao@uni-koeln.de

routes for the east-to-west expansion. Banks et al.[17] and Badino et al.[18] argued that climate change has been a main driver for the AUR dispersal. Shao et al.[19] analysed the best potential paths[20] for the AUR dispersal and largely confirmed the routes proposed by Mellars[16]. However, human dispersal is a highly convoluted process of advancement, retreat, abandonment and resettlement on different temporal and spatial scales. Palaeogenetic data with significantly higher resolution than technological data of the lithic inventories give perspective on this complexity, especially the pre-Aurignacian dispersal of AMHs into Europe. While IUP data indicate AMH dispersal impulses that reached Eastern Europe, but these did not lead to a significant settlement, in contrast to the AUR. The number of generations since the last common admixture event with NEAs is estimated to be about six to eight for Bacho Kiro and Oase[3]. For Zlatý kůň, the estimate is about 70 generations. This time window of ~200–2000 years is at the limits of the resolving power of paleogenetic but shows that population changes are as rapid as climate changes on century-to-millennial time scales.

The AUR was accompanied by major climate changes on millennial time scales, as reflected in the Greenland Stadial (GS) and Interstadial (GI) cycles highlighted by the Heinrich events (HEs) and Dansgaard–Oeschger events (D-Os)[21]. The AUR-P1 fell to the period between HE5 and HE4, while AUR-P2 between HE4 and HE3. The significant differences in the distributions of the archaeological sites attributed to AUR-P1 and AUR-P2 point to the temporal changes in population patterns. So far, our understanding of the techno-complex is mainly based on findings of archaeological excavations. Such data are essential in providing localised insight of an extremely complex picture, but they lack the power for macroscopic and quantitative interpretations of the population dynamics. They are insufficient for answering the basic question how the constellation of climate change and human activity influenced the human dispersal, demography and cultural evolution. Our view is that human dispersal must be considered as the manifestation of the human system which is, in terms of its internal dynamics, characterised by a large degree of freedom and a range of non-linear interactions on different scales, and in terms of external drivers, subject to unsteady and stochastic forcing. To provide quantitative answers to our questions, it is necessary to develop a human dispersal model framework which accounts for the biological, cultural, and environmental dimensions of the human system and integrates human- and natural science theories and diverse data types.

Human dispersal modelling has been active since the 1960s[22–25]. For large-scale problems, diffusion/reaction models are commonly used[26–28]. For small-scale problems, agent-based models are popular[29]. With the advancement in Global Climate Models (GCMs) and their paleoclimate applications, new models emerged by coupling large climate and archaeological data sets for studying climatic impacts on human dispersal[30,31]. By using a transient GCM simulation for the Pleistocene, Timmermann et al.[32] studied the spatiotemporal habitability for five hominin species over the past two million years and reported that astronomically forced changes in temperature, rainfall and terrestrial net primary production (NPP) influenced species distributions. Beyer et al.[33] reported that there are special climate windows for human dispersal if a threshold of 90 mm annual rainfall is breached for several decades. The use of NPP as a main driver for human mobility in earlier studies has advantages but also two deficiencies: (1) NPP is a poorly constrained quantity in GCMs; and (2) human existence not only depends on NPP but also on its accessibility and availability.

Here, we provide a detailed reconstruction of the AUR dispersal in Europe using a novel human dispersal model, the Our Way Model (OWM), combined with archaeological and paleoclimate data. In OWM, human existence potential (HEP) is used as a unifying quantity that drives the human population dynamics. HEP encompasses the various processes of the human system and has three layers of information, namely, the climate/environment HEP $\Phi_E$, accessible HEP $\Phi_{Ac}$, and available HEP $\Phi_{Av}$. While $\Phi_E$ measures whether climate/environment conditions are suitable for human existence and resources exist, $\Phi_{Ac}$ measures the human technological and cultural capacity to harness resources. The quantity $\Phi_{Av}$ measures whether the resources are available to individual humans and unifies the various mechanisms responsible for human mobility, not only climatic/environmental but also cultural and societal. The OWM is applied to simulating the dispersal of the AUR in Europe for a period of 20 kyr (45–25 ka) to provide a numerical reconstruction of the AUR dispersal history on the pan-European scale, which has not been possible before. Based on the model results, we quantify the spatial and temporal variations of the population density and fluxes, and interpret the AUR dispersal in light of the constellation of climate change and human exploration. A numerical chronology of the AUR dispersal in Europe is provided, which AUR verified for the locations where archaeological dating records exist. Insights into some of the highly disputed hypotheses, e.g., human dispersal routes, are provided. Note that while the OWM is general, the model parameters used here are specific for the AUR and the HEP data are produced based on the archaeological sites attributed to the AUR. Hence, the model results presented here are also specific for the AUR. However, for simplicity of description, expressions such as human dispersal, human population density etc. are used. It is clear that, unless specifically defined, such expressions refer to the AUR.

## Results and discussion

The OWM is run in the domain of (15°W–49°E, 20°N–60°N) with a spatial resolution of ~0.5° on the same numerical grid as for the climate data used for the HEP calculations. The simulations are done for various time periods, but the run for 45–25 ka with a numerical time step of 10 days is presented here (and additional runs are presented in the Supplementary Discussion). Here, we focus on the results for AUR-P1. For the simulation, HEP is first estimated for the prototypical stadial and interstadial conditions of the Last Glacial Period by combining GCM-simulated climate data and archaeological site data[21,34]. To enable a continuous OWM run for the study period, the estimated HEP for the prototypical stadial and interstadial conditions are interpolated/extrapolated in time based on the GICC05 time series of the NGRIP ice cores[35]. Since the precise origin of the AUR is still unclear, we follow here the hypothesis of Mellars[16] and take the area between the Levant and Dead Sea as the starting point of our simulation at 45 ka. The model results presented later and the sensitivity tests to the starting point confirm that this choice is plausible.

One aspect left out of the simulation is the possible interactions of the AMHs with the indigenous NEA inhabitants in Europe. Higham et al.[36] suggested that the Mousterian ended by 41–39 ka across Europe and the succeeding transitional industries, one of which linked with CHÂT, ended at a similar time. This indicates that the disappearance of NEAs occurred at different times in different regions and a significant overlap of 2600–5400 years might exist between them and the early AMHs in Europe (e.g. the Uluzzian). Hajdinjak et al.[3] showed that mixing between the NEAs and AHMs took place ~45–43 ka, prior to the appearance and spread of the AUR. Although the interactions between the indigenous inhabitants and AMHs cannot be ruled out, we assume here that the AUR expansion was unimpeded by the indigenous groups, but mainly determined by the climate/environment conditions and its own population dynamics. One of our studies[37] on the Iberian Peninsula shows that the end of the Middle Palaeolithic is indeed accompanied by a decline in the NEA population. In northern Italy[38] and Central Europe, stratigraphic evidence also points to a similar decline. These observations imply that the likelihood of NEA-and-AMH interactions decreases further west in Europe. Because of the large uncertainties associated with the potential interactions between the different populations, we decided not to include the interactions

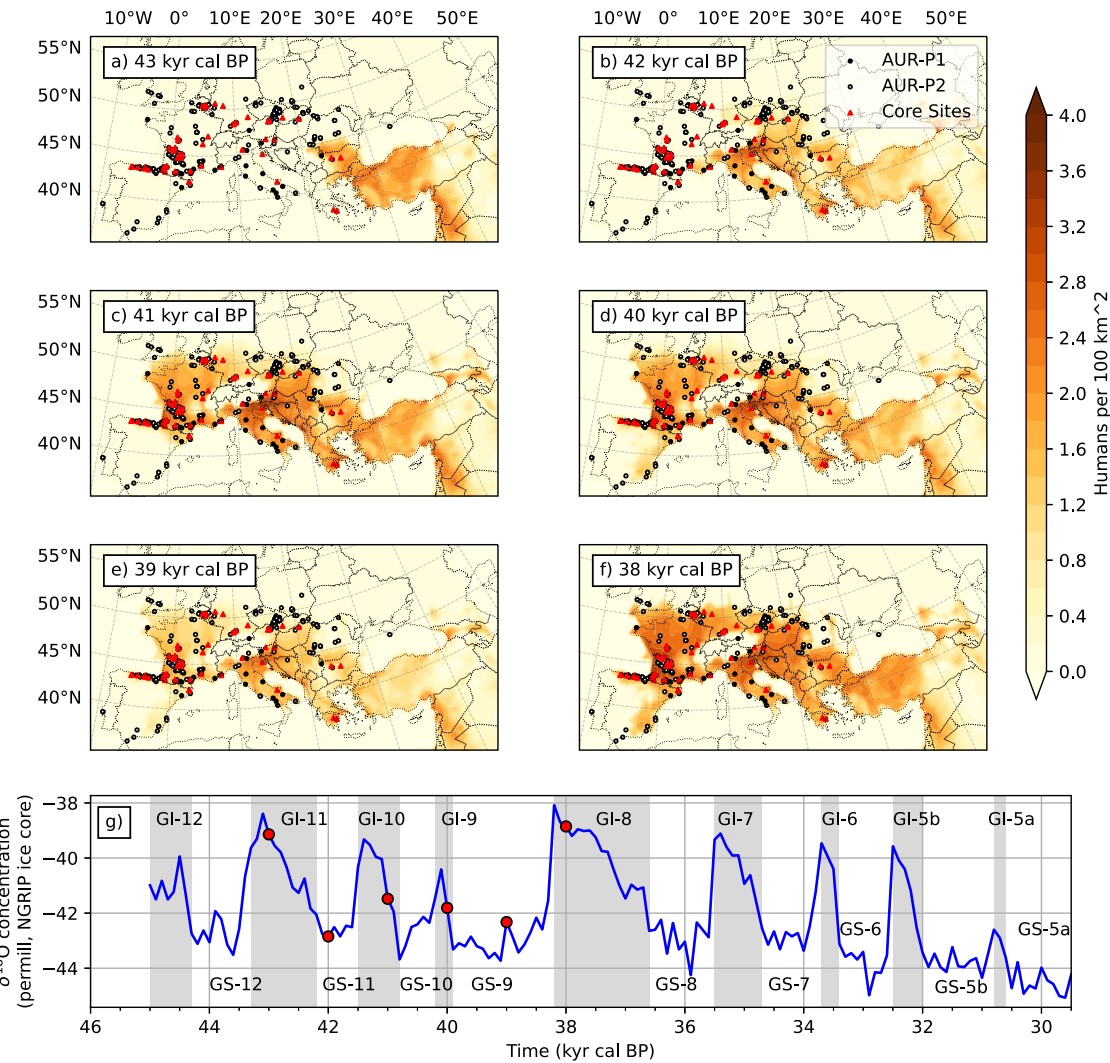

**Fig. 1 | Time slices of simulated population density.** The distributions of density in (P 100 km⁻²) for **a** 43, **b** 42, **c** 41, **d** 40, **e** 39 and **f** 38 ka are shown. The symbols represent the AUR archaeological sites for different phases (full black dots: Phase 1; open circle: Phase 2; and red triangles: Core sites). The $\delta^{18}$O time series is shown in (**g**). The times, for which population density is shown, are marked with full red dots. Shaded Areas correspond to interstadial (GI) and non-shaded stadial (GS) times.

between groups of different techno-complexes or hominin types in our simulations, as we would otherwise need more data and make speculations regarding the interactions (coexistence, assimilation, extermination etc.). Without these, even though group interactions can be formulated mathematically, reliable outcomes are unlikely.

Figure 1 shows the time slices of the simulated population density in Europe at 43, 42, 41, 40, 39 and 38 ka. Based on these results, the AUR human dispersal in AUR-P1 can be divided into the four stages of exploration, expansion, retreat and recovery, referred to as S1, S2, S3 and S4, respectively:

(S1) Exploration: in the first 2000 years ( ~45–43.25 ka), the west-ward expansion of AUR humans was relatively slow, due to rugged topography of the Anatolian Plateau, the unfavourable stadial conditions of the GS12 around 44 ka, and the initially low population density which exerts an attraction to more populated areas. By 43 ka (Fig. 1a), the west frontier of the AUR population arrived at the Balkan region.

(S2) Expansion: the next two thousand years ( ~43.25–41 ka) saw a rapid expansion of AUR humans to western Europe (Fig. 1a–c). This rapid expansion was related to the overall increase in population density and the generally warm climate conditions of the GI11 ( ~43 ka) and GI10 ( ~41 ka), interrupted only shortly by the GS11 which lasted several hundred years around 42 ka. By ~41 ka, the

maximum extent of the settlement as documented by the AUR-P1 archaeological sites was reached, with high population regions in southwestern France, northern Italy and Slovenia (Fig. 1c). An estimated maximum population of about 60,000 for AUR-P1 was reached, as will be further discussed later. Highham et al.[13] reported that the earliest AMHs appeared in Europe around 43–42 ka, and Mellars[14] reported that AMHs then rapidly expanded in Europe as the apparently sudden flowering of the Upper Palaeolithic suggests. The model prediction is consistent with these existing views on the sudden appearance of the techno-complex in Europe based on archaeological findings and confirms that the rapid expansion took place in interstadial time.

(S3) Retreat: in the next two thousand years (41–39 ka), Europe experienced two successive stadial periods (GS10 and GS9/HE4), interrupted only shortly by GI9 which lasted about 200 years around 40 ka. The GS9/HE4 period lasted for almost 1500 years. As a consequence, the total population in Europe significantly reduced and the size of the settlement areas shrank (Fig. 1e; see below), in particular in northern Europe. It is however also noted that while the population density generally decreased, humans largely survived the prolonged stadial conditions and relatively high population density sustained in the climate shadows of large topographic fea-tures, such as the area to the southeast of the Alps.

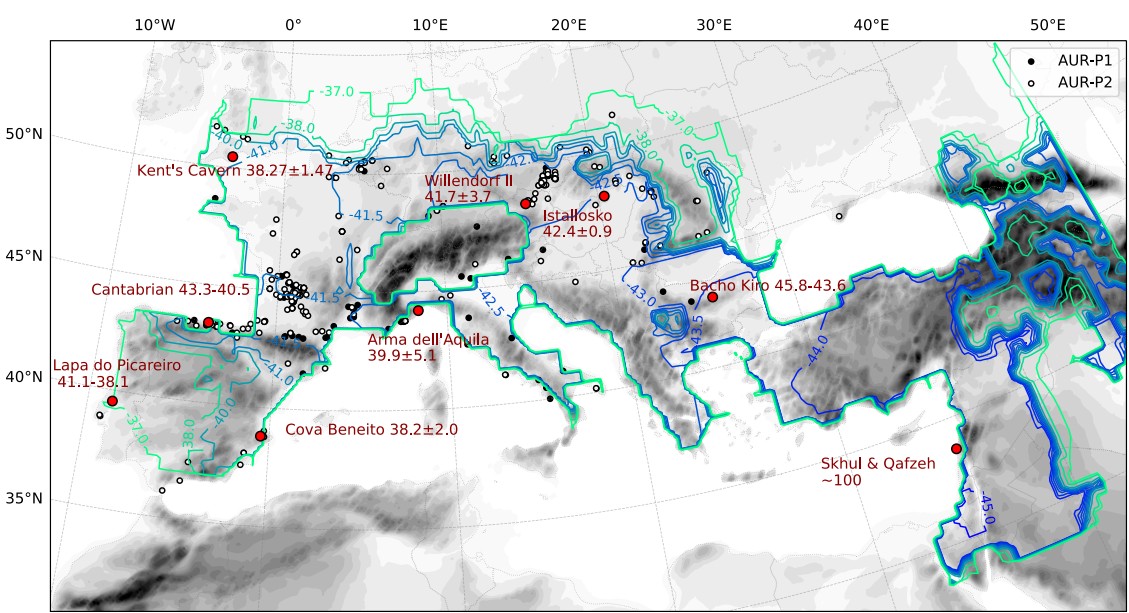

**Fig. 2 | Contours of the arrival time of human expansion in ka, defined by population density reaching 0.4 P 100 km$^{-2}$ for the first time.** Plotted are also the archaeological sites as explained in Fig 1. The red dots are the archaeological sites (Bacho Kiro, Istállós-kö, Willendorf, Arma dell Aquila, Cantabrian, Cova Beneito and Kent's Cavern) where $^{14}$C data (numbers shown) are available for comparison.

(S4) Recovery: the AUR population declined to a minimum at ~39 ka, but experienced a sharp increase shortly before 38 ka, marking the beginning of AUR-P2. In the post-GS9/HE4 era, the population quickly recovered in the next two thousand years, as seen in the large increase in population density and expansion into previously empty spaces, including southern Iberia, northern central Europe, the southern part of the British Isles, and northern coastal area of the Black Sea. By 37 ka, the maximum extent of the settlement as documented by the AUR-P2 archaeological records was reached (not shown).

The maximum settlement area of AUR-P2 clearly exceeded that of AUR-P1 prior to GS9/HE4. Our model does not yet contain explicitly a component for human adaptation to climate change, but a comparison of the climate conditions in the AUR-P1 and AUR-P2 settlement areas reveals that the AUR population that survived the challenges of the GS9/HE4 was in AUR-P2 better adapted to cold climate conditions[39]. Shao et al.[21] showed evidence for such an adaptation by comparing the probability density functions (PDFs) of the bioclimate variables at the AUR-P1 and AUR-P2 archaeological sites. They reported that a main peak of Bio1 (annual mean temperature) at ~6 °C and Bio4 (temperature seasonality) at ~7 °C exists in both AUR-P1 and AUR-P2, indicating that humans in both phases preferred to settle in relatively warm regions with moderate seasonal temperature variations. For AUR-P2, however, an additional secondary peak exists at Bio1 ~ 2 °C and Bio4 ~8 °C. This implies that in the post-GS9/HE4 era, a proportion of the population was better adapted to a climate of much lower annual mean temperature with higher seasonal temperature variability. Areas of such climate conditions, which were not settled in AUR-P1, became settled in AUR-P2, shifting the northern boundary of the AUR settlement areas northwards by two to three and up to five degrees of latitude in the post-GS9/HE4 era. The results shown in Fig. 1 are logically plausible and qualitatively in good agreement with the archaeological data.

Using the model-predicted population density, the chronology of AUR human dispersal can be reconstructed (Fig. 2). The human dispersal front is defined here by the locations where the population density reached 0.4 P 100 km$^{-2}$ for the first time. Fig. 2 shows that after

an initial northward expansion from the Levant, human dispersal was slowed down by the Anatolian Plateau and the stadial conditions of GS12. By ~43 ka, the dispersal front reached the Balkans. The AUR humans then took the advantage of the favourable interstadial climate conditions of GI11–GI10 and quickly expanded in western Europe. Additional expansion became difficult after ~41 ka and the outer boundary of the settlement areas remained either unchanged or somewhat contracted during the next 3000 years. In the post-GS9/H4 era, northward expansion reassumed in northern Europe and southwestward expansion on the Iberian Peninsula.

The model-predicted chronology of the first arrival time can be compared with the $^{14}$C dates of a number of AUR sites. For this purpose, an ensemble of simulations with perturbed key model parameters is generated and the ensemble mean arrival time and standard deviation for these sites are calculated as listed in Table 1. According to Fewlass et al.[40], the Bacho Kiro site is dated to 45.8–43.6 ka and the model-predicted first arrival time is 43.7 ± 0.37 ka. The model-predicted first arrival times at Istállós-kö, Willendorf are also consistent with the available $^{14}$C data. At the western end of Europe, a series of Cantabrian sites has been dated to 43.3–40.5 ka[41], in good agreement with the model prediction of 41.6 ± 0.67 ka. At the Mediterranean coastal sites Arma dell Aquila and Cova Beneito, model and observation are consistent within the error margin. According to

### Table 1 | Comparison of model-predicted arrival time and 14C dating of archaeological sites

| Site | Country | Lon (E) | Lat (N) | $^{14}$C (ka) | Model (ka) |
|---|---|---|---|---|---|
| Willendorf II[70] | Austria | 15.3990 | 48.3230 | 41.7 ± 3.7 | 42.6 ± 0.53 |
| Kent's Cavern[42] | U.K. | −3.530 | 50.470 | 38.27 ± 1.47 | 40.6 ± 0.65 |
| Istállós-kö[71] | Hungary | 20.3828 | 48.1189 | 42.4 ± 0.9 | 42.8 ± 0.52 |
| Cova Beneito[72] | Spain | −4.66 | 38.7981 | 38.2 ± 2.0 | 41.5 ± 0.66 |
| Bacho Kiro[40] | Bulgaria | 25.4167 | 42.9333 | 44.7 ± 1.1 | 43.7 ± 0.37 |
| Arma dell'Aquila[73] | Italy | 8.3245 | 44.1989 | 39.9 ± 5.1 | 42.5 ± 0.55 |
| Cantabrian[41] | Spain | −10.918 | 43.796 | 43.3 − 40.5 | 41.6 ± 0.67 |
| Lapa do Picareiro[43] | Portugal | −12.000 | 39.641 | 41.1 − 38.1 | 37.0 ± 1.00 |

Higham et al.[42], the Kent's Cavern site, dated to $38.27 \pm 1.47$ ka, is attributed to the AUR techno-complex that emerged on the British Isles slightly later than 40 ka. For this part, the model predicted first arrival time is around $40.6 \pm 0.65$ ka, much earlier than the [14]C data suggest. Note that at the fringes of the human settlement, the gradient of the first arrival time is large, and hence the model-predicted and [14]C-dated first arrival time in these areas are expected to show larger discrepancies.

A detailed analysis on the HEP for both the NEA and AMH populations[37] shows that it is unlikely that the two populations extensively coexisted on the Iberian Peninsula and hence unlikely that the possible presence of the NEAs significantly affected the expansion of the AMHs there. The model simulation (Fig. 1) reveals that the slow expansion of the AMHs into southern Iberia is attributed to the constellation of the dispersal process and climate change. At the time of GI10, the AMHs just arrived in northern Spain via southern France but were mostly settled in the Cantabrian coastal areas. The migrants probably did not expand southward to cross the Ebro Valley at the time, which was the northern boundary of a large territory south of the valley that was hostile to the AMHs in AUR-P1. The AMHs arriving northern Iberia were soon facing the prolong cold phase of GS10-GS9/HE4, which lasted almost 3000 years. On the pan-European scale, the AMH population was on the retreat (see below), and the conditions for AMH existence on the peninsula were also deteriorated. Thus, during the entire GS10-9/HE4 period, there was little chance for the AMHs to expand southward. Only after the GS9/HE4, in AUR-P2, the AMHs headed southward along the coast in today's Valencia and Andalucía, while the interior of Iberian remained unsettled.

However, Haws et al.[43] reported that the Lapa do Picareiro site in Portugal was populated by humans of the AUR as early as the first settlement phase (AUR-P1), around 41.1–38.1 ka. If true, their findings would imply that the spread of AUR humans in Iberia was much faster than previously known. The OWM simulation suggests that the AUR humans did not arrive the Lapa do Picareiro area until around 37.0 ka. From a macroscopic perspective, the Lapa do Picareiro site appears to be an outlier in the AUR-P1 data set, as it was separated from all other AUR-P1 sites and it is difficult to interpret how AUR humans appeared there in the first settlement phase of the AUR. A recent study by Klein[44] based on high-resolution (10 km) climate data shows that a potential route weakly linking Lapa do Picareiro with the Franco-Cantabria area via the western part of Spain could have existed. This suggests that an early human arrival at Lapa do Picareiro was not an impossible event, but one of low probability. The OWM simulation reported here, focusing on the pan-European scale, was obtained with a spatial resolution of 50 km. A study with a higher resolution is being prepared to reconcile the discrepancies between the model results and the findings of Haws et al.[43] surrounding this highly debated site[45].

To quantify the overall demographic changes in Europe, we define the total population $\Lambda$ and a mobility index $M$ as

$$\Lambda = \sum_{i,j} \rho_{i,j} s_{i,j} \qquad M = \sum_{i,j} |\mathbf{f}_{i,j}| \tag{1}$$

where $\rho_{i,j}$, $s_{i,j}$ and $\mathbf{f}_{i,j}$ are population density, cell area and population flux at the grid cell $(i,j)$, respectively, and the summation is over all grid cells in the study domain. As the numerical grid is fixed, $s_{i,j}$ are constant but differ slightly for different cells due to grid distortion. The model predicts $\rho_{i,j}$ and the total population in the simulation domain $\Lambda$ is thus the sum of $\rho_{i,j} s_{i,j}$ over all grid cells. The quantity $M$ represents the overall strength of human mobility.

Our model simulations show that climate change had a significant impact on the size of the population and human dispersal during the AUR. This observation is consistent with the assumptions made in numerous earlier studies. Figure 3a shows the climate and

archaeological timelines of the AUR and Fig. 3b time series of the predicted total AUR population $\Lambda$ and mobility index $M$. During the stages of S1 and S2 from 45 to 41 ka, $\Lambda$ linearly increased from about 5000 to a maximum of $\sim 60,000$. Existing meta-population estimates for the AUR based on the distribution of archaeological sites have large uncertainties. For western and central Europe Schmidt and Zimmermann[12] reported a population size of about 1500 humans with an interval of [800, 3000], while for Europe Bocquet-Appel et al.[46] reported a population size of about 4000 with a 95% confidence interval of [1700, 37000]. Our model-estimated $\Lambda$ maximum is about two times that upper limit[46]. The main reason for the discrepancies lies in the high-hep-no-site areas (i.e., areas of high human existence potential but with no archaeological sites found), for which our model may predict human presence while demographic estimates based on primarily on archaeological data assume zero or very low human density.

The model-predicted strong initial growth in $\Lambda$ was mainly attributed to the expansion of the settlement areas and the increase in population density, although the climatic impact was also recognisable. In the first 500 years, human mobility was confined to the Levant, as further dispersal was hindered by the rough terrains there and the onset of GS12. Associated with the GI11, a short burst of dispersal occurred, with the expansion front reaching the Balkan and Rhodope mountains. The rapid expansion stage, started shortly before 43 ka, was accompanied by a significant increase in population size and mobility. Beyond this point, $\Lambda$ was well-correlated with climate change, with a time lag of several hundred years. The mobility index $M$ was also well-correlated with climate change. The peak values of $M$, representing strong human mobility, occurred repeatedly in correspondence to Dansgaard–Oeschger events.

The AUR population experienced a major setback during the GS10-9/HE4 period, as the total population $\Lambda$ decreased from $\sim 60,000$ at 41 ka to about 35,000 at 39 ka (with a short recovery during GI9 around 40 ka). This corresponds to a 40% drop in population over 2000 years. This decline is attributed to both the decrease in population density and the shrinking of settlement areas, e.g., the northern rim of the settlement was contracted to lower latitudes. In the post-GS9/HE4 settlement phase, starting shortly before 38 ka, the total population quickly recovered. The maximum of $\lambda$ for AUR-P2 exceeded that for AUR-P1, reaching 80,000 at $\sim 37$ ka. As Figs. 1f and 2 show, both the population density and the settlement areas increased, e.g., the outer borders shifted further north and expanded into Iberia. The model suggests that it is probably around this time when the AUR humans reached Lapa do Picareiro. During AUR-P2, the population size remained relatively steady, showing much weaker reactions to climate change, as humans became better adapted to colder climate conditions.

Population flux $\mathbf{f} = \mathbf{v}\rho - D\nabla\rho$, consisting of a drift term $\nabla\rho$ and a diffusion term $-D\nabla\rho$, can be used to study population networks and dispersal routes. It describes the number of humans moving across a section of unit distance per time. Figure 4a shows, using the streamlines of $f$ averaged over the period of 44–41 ka (including stages S1 and S2), how humans dispersed into Europe in a major event. The model-predicted main human dispersal routes confirm the existence of the Danube and Mediterranean-Coast routes as hypothesised by Mellars[16]. An outstanding feature of the dispersal network is the human dispersal highway of large population fluxes, running almost parallel to the 45°N latitude circle and stretching $\sim 4000$ km from the Anatolia via the Balkan Peninsula, northern Italy, southern France to northern Cantabria. It is tentative to divide the dispersal network into four main routes, namely, the Balkan, Danube, Mediterranean-Coast, and West-Europe routes. The Balkan route joints the Anatolian area with the Danube routes south of the Balkan Mt. As the main dispersal stream continued westward along the Danube route, northward dispersal occurred crossing the Wallachian Plain around southern Carpathian

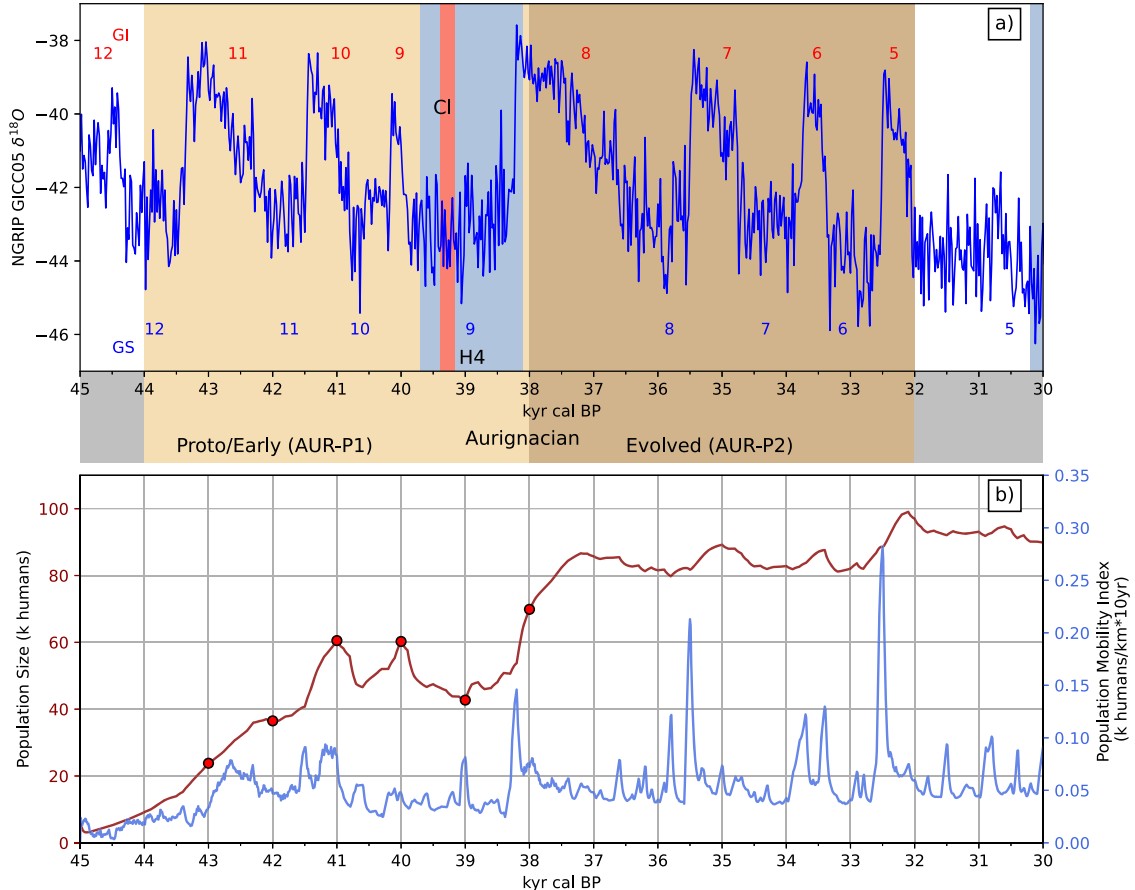

**Fig. 3 | Ice core time series, population size and population mobility index.**
**a** Time series of the NGRIP GICC05 $\delta^{18}O$ as a surrogate for temperature (larger values corresponding to warmer conditions) and the approximate archaeological timelines of the AUR. In Europe, the techno-complex developed in the late half of the Last Glacial Period, ~44–32 ka, when climate was featured by Greenland Stadial (GS, blue numbers) and Interstadial (GI, red numbers). The abbreviation CI stands for Campanian Ignimbrite volcanic eruption. **b** Time series of the model-predicted total AUR population $\Lambda$ and population mobility index $M$ for the period 45–30 ka. The red dots correspond to the time slices, for which the patterns of population density are shown in Fig. 1.

into the west Pannonian Basin. At the southwest of the Alps, Danube and Mediterranean-Coast routes merged and then diverged, with the mainstream continuing its westward journey and the other along the Danube route around the Alps to the east Pannonian Basin and western Europe. At southern France, the Mediterranean-Coast further diverged, with one branch continuing its westward expansion to Cantabria and the other following the West-Europe to the Paris Basin in northern France and from there, together with the humans arrived in Germany via the Danube route, further to the British Isles. The predicted dispersal network is fully consistent with the distribution of the archaeological sites attributed to AUR-P1.

Humans dispersed northward from southern France along a minor route through the Rhone valley. Vanhaeren and d'Errico[47] reported a geographic cline in ornament-type association sweeping counter-clockwise from the northern plains to the Eastern Alps through western France, northern Spain, the Pyrenees and Mediterranean Europe. This counter-clockwise sweeping cline is consistent with the modelled dispersal network. The latter authors also reported a marked contrast between the extremes of this cline, Germany and Austria, which, in spite of their geographic proximity had no ornament types in common. This contrast can be explained using the model results which show that Germany and Austria were located in different dispersal routes.

Once Europe was almost settled after the main expansion wave, the dispersal can be better characterised by a back-and-forth movement at the settlement fringes, as a response to climate change. Figure 4b shows the contours of population density $\rho = 1\,P\,100\,km^{-2}$ for

40, 39 and 37.8 ka. With the onset of the GS9/HE4 event, from 40.0 to 39.0 ka, the highly populated areas clearly contracted, especially in northern Europe and on the Iberian Peninsula, but in the post-GS9/HE4 era, high population areas substantially expanded northward in northern Europe and southward in Iberia.

## Methods
### Model framework
We model human dispersal as a manifestation of the human system which has biological, cultural and environmental dimensions. The processes on these dimensions interact, causing complex feedbacks and producing population density change in space and time, i.e., the human dispersal. The basic drivers on the biological dimension include innate responses, physiological adaption, and cognitive evolution, and on the cultural dimension social organisation and progress in technology and science, language and literature, religion etc. These processes occur on a spectrum of scales. The scale-transgressive nature is also obvious for the environmental dimension, e.g., climate and vegetation change on scales from seasonal to orbital. While nature offers a large amount of resources, what can be assessed is limited by the human capacity to harness them. In a population cycle, periods of rapid population increase and/or expansion correspond to the constellation of favourable environmental conditions and the accumulative increase in human capacity to use resources.

A framework is proposed here to incorporate all three dimensions to quantitatively model human dispersal. It consists of the components of paleoclimate/environment data and archaeological data, the

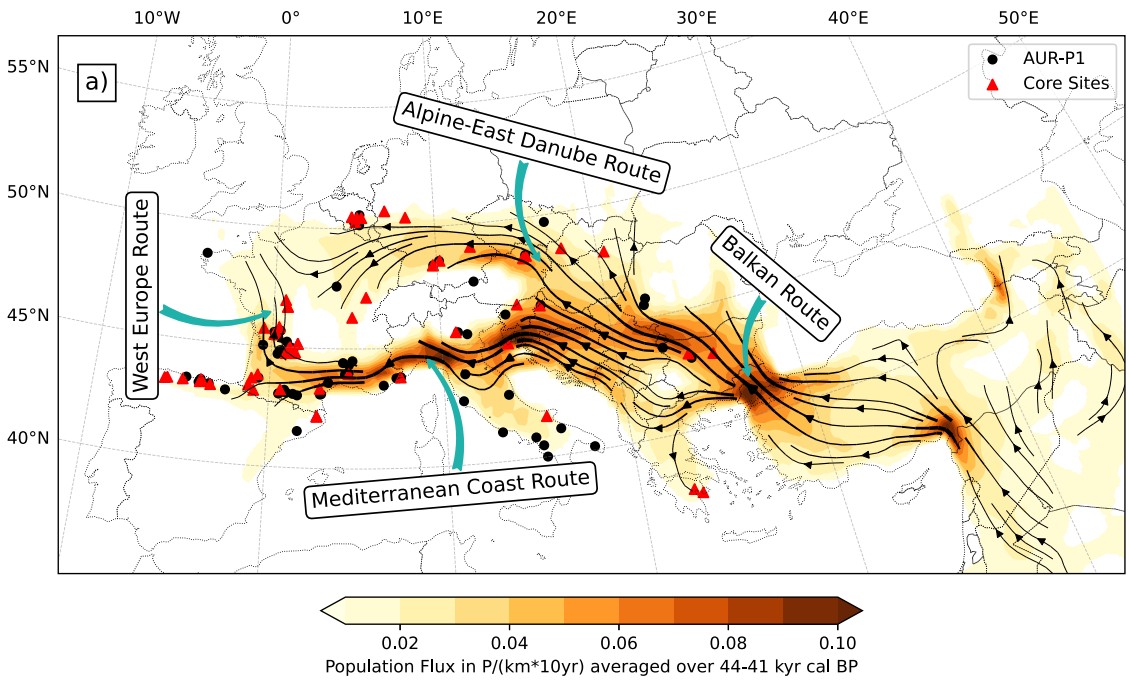

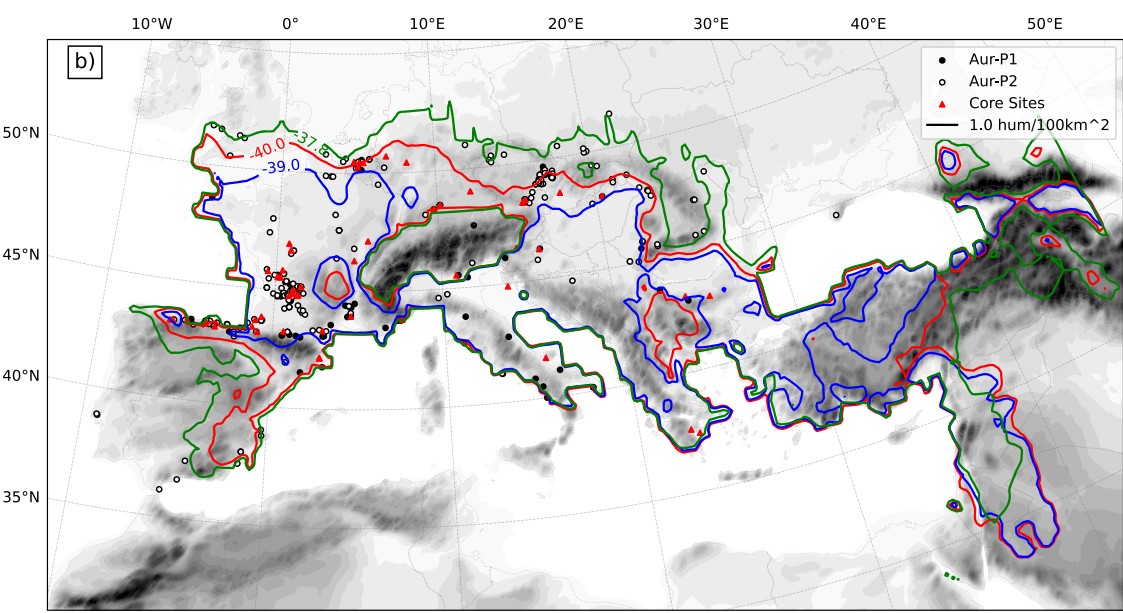

**Fig. 4 | Dispersal routes and population extend. a** Population flux in (P km⁻¹ 10 yr⁻¹) averaged over the period of 44–41 ka and the main human dispersal routes, including the Balkan, Alpine-East-Danube, Mediterranean-Coast and West-Europe routes. **b** Contours of $\rho = 1$ P 100 km⁻² for 40, 39 and 37.8 ka showing the extend of occupation. Today's topography is shaded grey.

HEP model and the human mobility model. We use HEP as a unifying quantity that drives the population dynamics. As Fig. 5 illustrates, HEP has three layers of information, including the environment HEP $\Phi_E$, accessible HEP $\Phi_{Ac}$, and available HEP $\Phi_{Av}$. $\Phi_E$ measures whether climate/environment conditions are suitable for human biological existence and whether resources are abundant. The input for computing $\Phi_E$ can either be independent (e.g., temperature and rainfall) or derived variables (e.g., net primary production) or both. To determine the preferential climate/environment conditions for human existence of a given culture, either knowledge a priori or information derived from archaeological data is required.

The quantity $\Phi_E$ sets the upper limits for human existence, but these limits are very loose because human existence is also restricted by the human capacity to harness resources and adapt to environmental changes. $\Phi_{Ac}$ is an extension of the cultural carrying capacity widely used in studies of population dynamics[48,49]. Note however that $\Phi_{Ac}$ here is not constant but a function of space and time, depending on climate/environment conditions and human activities. $\Phi_{Ac}$ describes the overarching effect of the cultural dimension of the human system, and embedded in it are processes such as technological progress, societal structure, cultural behaviour, religion etc. Thus, the estimation of $\Phi_{Ac}$ is not straightforward, but relies on the parameterisation of such processes identified in anthropology and the use of machine learning from archaeological data[34].

Regions of high $\Phi_{Ac}$ are attractive to humans, where the population will likely grow until a critical level is reached. Beyond this level,

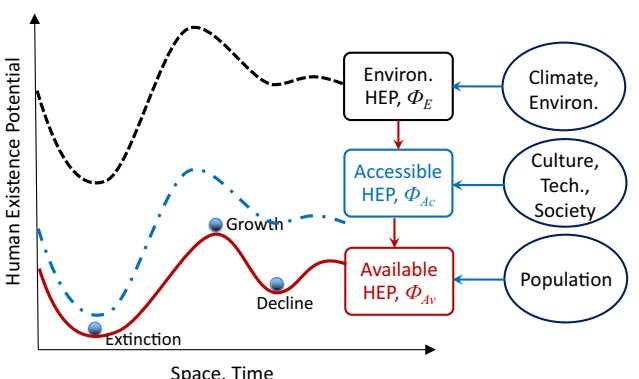

**Fig. 5 | HEP framework.** Schematic illustration of relationships between environment, accessible and available HEP and available HEP as a driver for human dispersal.

the availability of resources per human becomes a limiting factor which exerts pressure to drive humans to more favourable areas or to reduce the production rate. Therefore, $\Phi_{Av}$ is both a driver of the population dynamics and a dependent on population density, highlighting the non-linear dynamics of the human system.

Further explanations and equations regarding the HEP are given in the Supplementary Methods.

## Human mobility model

We define human dispersal as the propagation of population density. For the Palaeolithic, the anthropogenic impact on the environment is neglected. The population density (commonly in persons per 100 km$^{-2}$, i.e., P 100 km$^{-2}$) satisfies

$$\frac{d\rho}{dt} = p \tag{2}$$

where $t$ is time and $p$ is net production rate (i.e., birth rate $b$ minus death rate $d$). Equation (2) predicts an exponential population growth if $p$ is linear to $\rho$. As both $\rho = \rho(x, y, t)$ and $p = p(x, y, t)$ are functions of space (x, y) and time, we rewrite Eq. (2) as

$$\frac{\partial\rho}{\partial t} + \mathbf{v} \cdot \nabla\rho - D\nabla^2\rho = p \tag{3}$$

where $v$ is drift velocity and $D$ is diffusion coefficient. In earlier studies[30], human dispersal is modelled as a diffusion/reproduction process, with no population drift. Here a drift term is included to better represent the macroscopic processes which systematically drive human dispersal. Equation (3) is capable of representing the various modes of human mobility known in demographic descriptions, e.g., the leapfrog mode[50], can be represented in Eq. (3) by the drift term. Also, the human dispersal routes may just be the consequence of strong diffusion in a preferred direction, e.g., in the direction of gradient $\Phi_{Av}$. Numerically, Eq. (3) is solved in two steps

$$\frac{\partial\rho}{\partial t} = -\mathbf{v} \cdot \nabla\rho + D\nabla^2\rho \tag{4}$$

$$\frac{\partial\rho}{\partial t} = p \tag{5}$$

on a two-dimensional (longitude x and latitude y) grid, subject to initial and boundary conditions.

The quantities $v$, $D$ and $p$, all functions of $\Phi_{Av}$, need to be specified, but the $\Phi_{Av} \sim \rho$ relationship needs to be established

beforehand. First $\Phi_{Ac}$ is scaled to 0 and 1. For a given $\Phi_{Ac}$, an optimal population density, or the local carrying capacity $\rho_c$, exists at which the resource availability per human is maximum and the population pressure is minimum. The maximum of $\rho_c$ (at $\Phi_{Ac} = 1$), i.e., $\rho_{c,max}$ is a quantity that is culture-dependent, or the cultural carrying capacity. In general, $\rho_{c,max}$ evolves slowly but changes suddenly in certain phases of the population cycle (e.g., cultural revolution). For the AUR, we specify $\rho_{c,max}$ as an external parameter or a linear function of time, and compute $\rho_c$ as

$$\rho_c = \rho_{c,max}\Phi_{Ac} \tag{6}$$

Two competing processes of the human system are considered in deriving $\Phi_{Av}$ from $\Phi_{Ac}$: (1) at low population density, mutual support is critical to increase the societal efficiency in accessing and utilising the resources; (2) once the population density exceeds a critical level, the resources available per individual become scarce, creating a population pressure. These two processes are conveniently represented using a function in the form of a Weibull distribution

$$w = \frac{\eta}{\epsilon}\left(\frac{\frac{\rho}{\rho_c}}{\epsilon}\right)^{(\eta-1)} \cdot \exp\left[\left(\frac{\frac{\rho}{\rho_c}}{\epsilon}\right)^{\eta}\right] \tag{7}$$

where $\epsilon$ and $\eta$ are scaling and shape parameters, respectively. We then express $\Phi_{Av}$ as

$$\Phi_{Av} = \frac{w}{w_{max}} \cdot \rho_c \tag{8}$$

Now $\Phi_{Av}$ is scaled from 0 to $\rho_{m}ax$ as $\Phi_{Ac}$ is scaled from 0 to 1. Figure 6 shows two examples of $\Phi_{Av}$ as function of $\rho$ for different $\rho_c$ values. With $\epsilon = 0.4$ and $\eta = 1.6$ in Eq. (7), $\Phi_{Av}$ reaches a maximum at $\rho = 0.2\rho_c$.

The drift velocity is related to the gradient of $\Phi_{Av}$ via

$$\frac{d\mathbf{v}}{dt} = \alpha\nabla\Phi_{Av} - \gamma\mathbf{v} \tag{9}$$

with $\gamma$ being a damping coefficient. Equation (9) represents a systematic drift of population in the direction of available HEP. The linear coefficient $\alpha$ is given by

$$\alpha = \frac{V}{M \cdot \tau} \tag{10}$$

where $V$, $M$ and $\tau$ are scaling drift velocity, scaling $\overrightarrow{\nabla}\Phi_{Av}$ and scaling response time, respectively. The diffusion coefficient D is expressed as

$$D = \kappa \cdot l \cdot \sigma_v \tag{11}$$

where $\kappa$ is a model parameter, $l$ is the typical distance and $\sigma_v$ the velocity variability (standard deviation) on the sub-grid scale. The diffusion term in Eq. (4) is an aggregated expression for two sub-grid processes: (1) it represents the mobility arising from the exploratory nature of humans. This process is typically stochastic and difficult to predict in detail, but its consequence is a diffusion of humans from areas of high population density to areas of low population density; (2) it represents the small-scale variability of the drift process. Within a grid cell, humans may systematically relocate to areas of more favourable conditions. Due to the limited model resolution, such sub-grid dynamics are not represented by the drift velocity but appear as diffusion. The diffusivity $D$ may be directional, such that diffusion in a certain direction may be faster. This, together with directed drift, mimics human dispersal along preferred routes.

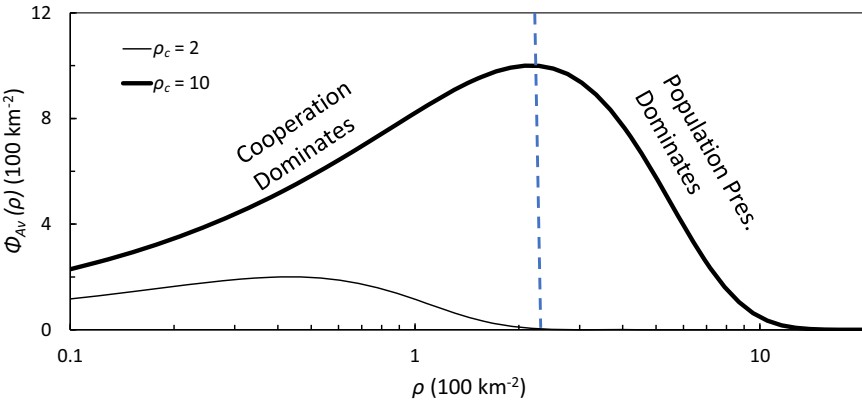

**Fig. 6 | Available HEP distributions.** Examples of $\Phi_{Av}$ as a function of $\rho$ for two different $\rho_c$ values. The parameters $\epsilon$ and $\eta$ in Eq. (7) are set to 0.4 and 1.6, respectively.

The diffusivity is parameterised by specifying the scale distance $l$ and scale velocity variability $\sigma_v$. It is assumed that

$$l = \kappa_l \cdot \nabla \quad (12)$$

$$\sigma_v = \kappa_\sigma \cdot |\mathbf{v}| \quad (13)$$

with $\kappa_l \sim 1$ and $\kappa_\sigma \sim 0.1$ being two empirical coefficients, $\nabla = \sqrt{\Delta x \Delta y}$ the typical grid resolution and $|\mathbf{v}|$ the magnitude of $\mathbf{v}$. We have carried out detailed investigations on the diffusion process using models in which the position and velocity of individual humans are explicitly modelled.

The reproduction rate $p$ is assumed to be a function of $\rho$ and $\Phi_{Av}$. This is a strong simplification of the reality. Other factors, which influence p, such as age profile, marriage rules and group interactions, are yet to be fully implemented in the model. As presented by Steele[24], $p$ is modelled using a logistic function,

$$p = r\rho \left(1 - \frac{\rho}{\rho_c}\right) \quad (14)$$

where $r$, which defines the growth rate, is a function of $\Phi_{Av}$ (e.g., $r = r_{max}\Phi_{Av}$). Equation (14) implies that as $\rho$ approaches $\rho_c$, p reduces to zero and for $\rho > \rho_c$, p becomes negative. For the AUR, $r$ is the order of 0.01 yr$^{-1}$. Schmidt et al.[51] showed that the estimates of $r$ and $\rho_{c,max}$ are very uncertain for the AUR and thus they are considered to be tuneable model parameters.

**Human existence potential**

The archaeological site data provide the information on where and when humans existed, and the climate/environment data at the sites provide the information on the conditions under which humans existed. These two data sets are combined to estimate the HEP. The archaeological data used here include 371 AUR sites in Europe[52] with well-documented, dated or diagnostic assemblages. The archaeological sites confined to Europe between 10°W and 30°E are used and the sites outside this area are excluded. The sites occupied in both AUR phases are referred to as "core sites" and those occupied only in one phase as "single-phase sites". In total, 60 core sites are identified. For AUR-P1, there are 118 sites, including 58 single-phase sites, and for AUR-P2, there are 312 sites, including 252 single-phase sites. Shao et al.[21] discussed the uncertainties in the archaeological database and the likely consequences.

Paleoclimate data are reconstructed for the study domain on a two-dimensional numerical grid with $I \times J$ cells and a resolution of 0.5° lat/lon, as detailed in the "Model Setup" section. The grid cell over large water bodies (e.g., ocean) are excluded from consideration. The grid cells where one or more archaeological sites exist are defined as "presence cells", and a subset of the grid cells where no archaeological site exists as "absence cells". To avoid possible stacked sites, a down-sampling procedure is applied to the training of the HEP model: only the presence records of the archaeological sites on the numerical grid are used instead of the archaeological sites directly, i.e., if a grid cell has more than one site, then this cell is treated only as one presence record. This treatment of the archaeological site data is equivalent to the spatial-blocking technique sometimes used for cross-validation of data with spatial structures[53], with the spatial-block size being the grid-cell size, namely, 0.5°. Roberts et al.[53] argued that spatial-blocking cross-validation provides a more realistic assessment of the model performance (here, the HEP model). The rest cells are "absence cells" which consist of "a priori absence cells" and "pseudo-absence cells". A priori absence cells are those where human presence was impossible according to our pre-knowledge. A set of criteria is used to define them, including (a) BIO1 (annual mean temperature) is below −2 °C or above 16 °C, and (b) BIO13 (rainfall in wettest month) is below 30 mm or above 250 mm. These values correspond to the 95% confidence level for human existence, estimated based on the probability density functions of the relevant variables at the archaeological sites. The rest absence cells are "pseudo-absence cells" which have three different interpretations: (1) regions humans existed, but no human records have been found (e.g., site not preserved); (2) regions humans could exist, but unexplored; and (3) regions unsuitable for human existence. Lobo et al.[54] referred to these pseudo-absence cells as "methodological", "contingent" and "environmental" absence cells. Supplementary Fig. 1a illustrates the spatial blocking of the archaeological site data and the definitions of a priori absence cells and pseudo-absence cells.

No general statement can be made about the nature of the pseudo-absence cells, but the following options have been considered in training the HEP model. (1) Pseudo-absence cells do not enter into the HEP model training, and the HEP model is trained using the presence and a priori absence records only; (2) Assuming the methodological, contingent and environmental absence are equally probable and 1/3 of the pseudo-absence cells enter into the HEP model training as absence cells and 1/3 as presence cells; (3) Assuming the probability of a finding archaeological record among the pseudo-absence cells is the same as that of finding an archaeological record reflected by the existing data set, namely, $N_p/(N_p + N_{pa})$, where $N_p$ is the number of presence records and $N_{pa}$ is the number of pseudo-absence records. In this study, we take option 3 which is the most stringent, resulting in a HEP model which tends to give smaller HEP values than the other two options. Some uncertainties are unavoidable here, but the HEP model is primarily determined by the presence and a priori absence cells.

It is not intended here to construct a HEP model that is entirely data-driven. Experiences show that models entirely data-driven may lead to outcomes which contradict common understanding. Thus, expert knowledge must be incorporated into the model a priori. Here,

this philosophy is reflected in the definition of the a priori absence cells. For example, it is extremely unlikely that Palaeolithic humans existed in polar regions and hence, the HEP there should be set to zero. The definition of the a priori absence cells is not only advantageous but also necessary for the logistic quadratic regression used in this study, as illustrated in Supplementary Fig. 2, for which cells with HEP equals zero must be defined.

The probability of human existence for the presence and absence (consisting of a priori absence cells and a subset of pseudo-absence cells, as discussed above) cells is assumed to be 1 and 0, respectively. We refer to the data set derived this way as the "presence/absence records". The HEP for a given cell and time $(x, y, t)$ is assumed to depend on a set of climate predictors. Using the presence/absence records and the predictors associated with them, a multi-variate second-order logistic regression model is constructed, namely, the HEP model. While different techniques can be used to derive the HEP model[54], we opted to take the logistic regression approach, as illustrated in Supplementary Fig. 2 because once the model is trained, it can be used to estimate the HEP at any $(x, y, t)$ where the predictors are known, without having to manipulate the predictors. In this way, the computation of HEP for thousands of years can be relatively easily done.

Seventeen bioclimatic variables[55] are taken as candidate predictors which are standardised according to $c_n = \frac{c - \bar{c}}{\sigma_c}$ with $\bar{c}$ and $\sigma_c$ being the mean and standard deviation of variable $c$. The selection the predictors requires (1) they are not collinear; and (2) allow the logistic regression to distinguish the presence/absence records. We use the hierarchical correlation cluster[34] technique to examine the collinearity between the candidate predictors. Shao et al.[21] showed that $T_m$ (annual mean temperature), $T_\delta$ (temperature seasonality), $R_m$ (precipitation of wettest month) and $R_\delta$ (rainfall seasonality) are the most suitable predictors for the Palaeolithic humans. These are also the main variables which determine the NPP, and thus, there is no need to explicitly use NPP as a predictor in the HEP model to avoid collinearity. We denote the selected predictors as vector $\vec{P}$, and construct a quadratic polynomial $q$ as

$$q(\mathbf{P}) = \frac{1}{2}\mathbf{P}^T \underline{A} \mathbf{P} + \mathbf{B} \cdot \mathbf{P} + C \quad (15)$$

where $\underline{A}$ is a matrix, $\mathbf{B}$ a vector and $C$ a constant, all to be estimated by training the model with the data. The HEP is mapped to $q$ via

$$\Phi_E(q) = \frac{1}{1 + \exp(-q)} \quad (16)$$

We train the model 1000 times, each with a random selection of 80% of the presence/absence records and use the rest 20% to evaluate the model performance. It has been shown[34,37] that the regression model gives good $\Phi_E$ estimates.

As shown in ref. 21 (and for completeness partially repeated here), we use the Brier Skill Score (BSS) as a measure of the model goodness, which is defined as

$$BSS = \frac{1}{N}\sum_{n=1}^{N}\left(1 - \frac{BS_n}{BS_0}\right) \quad (17)$$

where $BS_n$ is the Brier Score[56] for the $n$th model run, defined as

$$BS_n = \frac{1}{J}\sum_{j=1}^{J}\left(H_{nj} - H_{oj}\right)^2 \quad (18)$$

with $J$ being the number of the 20% presence/absence records, $H_{nj}$ and $H_{oj}$ are the model-predicted HEP and the presence/absence record at grid cell $j$, respectively. $BS_0$ is the Brier Score of the same run with $\underline{A}$

and $\mathbf{B}$, but $C$, in Supplementary Eq. (15) set to zero. Because

$$\Phi_E(q) = \frac{1}{1 + \exp(-C)} \quad (19)$$

is the simplest model, a BSS = 0 implies that the HEP model performance is identical to the simplest model, while a BSS = 1 points to a "perfect" model meaning that all training runs perfectly reproduce the 20% presence/absence records. A negative BSS is possible, indicating that the HEP model performance is worse than the simplest model. Another measure for the model performance is the "Area under a Receiver Operating Characteristics Curve". A large "Area under Curve" represents a high rate of correct classification by the model with respect observations[34]. The $N$ training runs allow an assessment of the model performance using the BSS and the Area under Curve.

As stated earlier, for the validation of the HEP model, the spatial-block cross-validation technique is used, with the spatial-block size being the same as the grid-cell size for the dynamic modelling of human dispersal. Because the existence of archaeological sites is correlated in space and time, it is appropriate to ask whether the spatial-block size is adequate to ensure the data for the cross-validation are independent of the data used for training the HEP model. We have conducted four additional tests using spatial-blocking sizes of 1.0, 1.5, 2.0 and 2.5°. The corresponding distributions of the presence, a priori absence and pseudo-absence cells are as shown in Supplementary Fig. 3. (A further run with a spatial-block size of 3.0° did not converge due to the very small number of presence cells left because of the spatial blocking and hence not shown) The performance scores of the HEP models trained using the different spatial-block sizes are listed in Supplementary Table 1. Based on the AUC and BSS values, we cannot tell whether the performances of these runs are significantly different from that of the base run with a spatial-block size of 0.5°.

The spatial-block size 0.5° is finally selected here for the following reasons. First, this size corresponds to the scale of core areas, a concept widely used in archaeology. Schmidt and Zimmermann[12] discussed the spatial scales embedded in archaeological data sets, ranging from local, catchment, core-area, supra-regional and global. The scale of core areas is a measure to distinguish intensively and probably continuously occupied areas from those that were either sporadically, marginally, or not occupied. Second, using a larger spatial-block size implies a smaller number of the presence records left for the model training (see Supplementary Table 1). Third, HEP is introduced to define the climate and environmental conditions under which humans existed, which are heterogeneous in space. The spatial-block size needs to be smaller than the dominant variation scale of climate and environmental conditions. Last, it is advantageous to set the spatial-block size to the grid-cell size for the HDM, as otherwise the HEP estimates must be again interpolated in space, causing additional errors.

The HEP model we finally used is the ensemble mean model of the 1000 training runs. A measure of the reliability of the HEP estimates is the ensemble standard deviation. As an example, Supplementary Fig. 1b shows the model-estimated for the AUR-P1 interstadial case and Supplementry Fig. 1c shows standard deviation. As seen, this quantity is mostly small, confirming the consistency of the HEP estimates. Relatively large values of the HEP standard deviation are found in several areas such as central Iberia and the northern border of the human settlement. For these areas, the interpretation of the HEP estimates warrants caution.

Certain environmental factors, such as water bodies, mountain terrains, forests etc., which affect human existence, are not explicitly used as predictors in the HEP model. For example, climate conditions over a large water body may be perfectly suited for human existence, but the area may be inaccessible for a continental techno-complex. Unlike bioclimatic variables, such environmental factors are almost

stationary on the time scale of cultural evolution and their impacts on human existence differ only for different cultures, i.e., areas inaccessible for one techno-complex may be accessible for another. In our study, these factors are accounted for by modifying the estimated $\Phi_E$ using additional modification functions to compute the accessible HEP. With the modifications, $\Phi_{Ac}$ is computed as

$$\Phi_{Ac} = \Phi_E \cdot g_{ele} \cdot g_{std} \cdot g_{wat} \cdot \ldots \cdot g_x \qquad (20)$$

where $g_{ele}$, $g_{std}$ and $g_{wat}$ are the modifications for terrain elevation, terrain-elevation standard deviation and water bodies, respectively. Additional modification functions $g_x$ can be assigned to reflect the human capacity to access resources. The modification functions are as set in ref. 19 but are still rudimentary and should be improved in future.

The AUR developed in the middle of the Last Glacial Period (LGP, ~115–11.7 ka) in which climate and environmental conditions underwent major changes. In Fig. 3, the NGRIP isotope $\delta^{18}O$ data are used as a surrogate for temperature to infer to the changes. In the second half of the LGP, it was featured by GS and GI cycles on millennial time scales, highlighted by extremely cold Heinrich events followed by extremely warm Dansgaard–Oeschger (D–O) events. The AUR experienced seven GS periods (GS12 to GS6), including a prolonged cold period of GS10-GS9/H4 period which lasted almost 3000 years ( ~41–38 ka), and seven GI periods (GI11 to GI5). The AUR-P1 falls to the GS/GI cycles marked by H5 and H4, while the AUR-P2 falls to the cycles marked by H4 and H3. With the profound climate changes and cultural evolution, major differences existed in the HEP for the different time periods of the techno-complex. However, as the dating of the archaeological sites come with large uncertainties, it is not possible to determine whether the sites were occupied under GS or GI conditions. In our analysis, we first estimated the HEP for AUR-P1 and AUR-P2 in prototypical GS and GI conditions, and then interpolated the HEP values for other times using the NGRIP ice core $\delta^{18}O$ values, as detailed in Section 3.3 Model Setup. For the prototypical conditions, we assumed that all archaeological sites existed in GI times and trained the HEP model accordingly. The trained HEP model was then applied to the GS times. As shown in ref. 21, this assumption is the most plausible, while the alternative, namely, that all archaeological sites existed in GS times would lead to results contradicting the common understanding (e.g., northward expansion of human settlement in stadial times). Supplementary Fig. 4 shows as an example the patterns of accessible HEP ($\Phi_{Ac}$) for prototypical interstadial and stadial times during the AUR. For the interstadial times (Supplementary Fig. 4a), most archaeological sites are found in the regions with HEP values larger than 0.5, with exceptions in the Iberia, southern Italy, Greece and the northern boundaries of human presence. Most of the core sites are in regions of high-HEP values, such as southwestern France and northern Spain.

It has been noted that some areas have high-HEP values but with no archaeological sites found, e.g., the south-eastern part of Hungary. These areas are referred to as High-Potential-No-Site (HPNS) areas, commonly called "empty" areas in archaeology. HPNS areas may occur for two different reasons: (a) they are non-settlement areas despite of the high potential or (b) they are settlement areas, but due to data bias, archaeological sites have not (yet) been found or are not preserved. A HPNS area not affected by data bias can be viewed as a potential expansion area. In this case, the absence of sites needs to be explained based on factors other than human existence potential, for example, they are yet to be explored by humans. Indeed, some of the HPNS areas for AUR-P1 were later settled by humans in AUR-P2, e.g., parts of northern France.

The northern border of human existence lies at about 50°N, beyond which, HEP rapidly decreases to the north and northeast, predicting a low probability of human existence there. The mountainous regions, such as the Alps, have low HEP values. For stadial times (Supplementary Fig. 4b), the HEP values are substantially reduced and

are larger than 0.5 only in isolated areas, mostly located in the climate shadows of large topographic features, such as northern Italy, the Rhone Valley, and the Carpathian Basin. These areas, apart from coastal Balkans, have evidence of human settlement, but with scarcely scattered core sites. While the climate conditions became more challenging for humans in central and southwestern Europe, human existence remains possible, with the HEP values at about 0.4. The HEP difference between interstadial and stadial times (interstadial-stadial) is shown in Supplementary Fig. 4c. In Europe, the HEP values in interstadial times are significantly higher, apart from the Iberian coast and Anatolia.

Note that Supplementary Fig. 4 only shows the potential for human existence, but whether humans existed at some point of space and time needs to be predicted using the dynamic component of the OWM. In summary, the OWM and its applications is schematically illustrated in Supplementary Fig. 5. The framework mainly consists of two components. The first component combines paleoclimate data and archaeological data using machine-learning techniques to estimate human existence potential, namely, the HEP model, and the second component include dynamic applications of two types of model simulations, the agent-based constrained stochastic model and the large-scale Human Dispersal Model (HDM) with ensemble simulations. Reported here are the results using the HDM simulations.

## Model setup

Simulations of the AUR human dispersal on the pan-European scale are carried out using the OWM for the period from 50 to 25 ka. For the two AUR phases, we use the prototypical stadial and interstadial times to represent the cold and warm extreme conditions, respectively. The Community Climate System Model version 4 (CCSM4[57]) is used to reconstruct the prototypic stadial and interstadial climate conditions in the Last Glacial Period, not the specific events. We then use the HEP model and the GCM-simulated climate data to calculate the accessible HEP $\Phi_{Ac}$ for the prototypical stadial and interstadial conditions in AUR-P1 and AUR-P2. For the GCM simulations, the atmospheric, ocean, land, and ice models of the CCSM4 are the Community Atmosphere Model[58], Parallel Ocean Program[59], Community Land Model[60] and Community Ice Code[61,62], respectively. The various data sets are unified to the numerical grid of the CCSM4 with a 0.5° by 0.5° horizontal resolution. For the HEP estimates and OWM simulations, the same grid and resolution are used. The boundary conditions and forcing for the CCSM4 simulations follow the PMIP3 and CMIP5 protocols for the LGM (https://wiki.lsce.ipsl.fr/pmip3/doku.php/pmip3:design:21k:final), but with modifications. The specifications of the ice-sheet extent, land-surface elevation and land-sea mask follow the PMIP3/CMIP5 LGM experiment. In a GCM without interactive glacier dynamics, an artificial forcing is required to model the paleoclimate stadial/interstadial excursions. To acquire a rapid weakening of the Atlantic Meridional Overturning Cell (AMOC) in stadial times, a surface freshwater flux anomaly of $0.25 \times 10^6$ m³ s⁻¹ is imposed, distributed uniformly over the northern North Atlantic (50°–75°N, 63°W–4°E). This corresponds to a net freshwater gain of 2.3 mm d⁻¹. The duration of the forcing is 100 years, a time span comparable to the duration of a Heinrich event (100–500 years). For stadial times, in addition to the freshwater forcing, the Atlantic sea surface temperature (SST) is decreased from the LGM SST by about 2K. For interstadial times, no extra freshwater forcing is applied, but the North Atlantic is assumed to be ice-free and the northern North Atlantic SST is increased from the LGM SST by about 5K. Using these model modifications, and the orbital parameters as set by Berger and Loutre[63], the CCSM4 model is initialised for 39 ka and run for 100 years separately for the interstadial and stadial conditions. The simulated air temperatures at 2 m height and the precipitation rates of the last 30 years of the 100-year runs are used as input variables of the HEP model. To enable a continuous simulation of the human dispersal over the entire AUR period, the HEP estimated for the

**Table 2 | Key parameters for the OWM simulations for the AUR dispersal on a pan-European scale**

| $\rho_{c,max}$ | Equation (4) | 5 P 100 km$^{-2}$ |
|---|---|---|
| $\epsilon$ | Equation (5), Weibull scaling | 0.4 |
| $\eta$ | and shape parameters | 2.5 |
| $\alpha$ | Equation (8), scaling parameter | 20 P km$^2$ yr$^{-2}$ |
| $\gamma$ | and damping coefficient | 0.1 yr$^{-1}$ |
| $D$ | Equation (10), diffusion coefficient | 10–100 km$^2$ yr$^{-1}$ |
| $r$ | Equation (12), reproduction rate | 0.01 yr$^{-1}$ |

four different situations are interpolated/extrapolated in time over the study period based on the GICC05 time series of the NGRIP ice cores[35]. The GICC05 time series consists of $\delta^{18}O$ values (as temperature proxies in 20-year time steps. For our purposes, the $\delta^{18}O$ values are normalised as

$$\delta^{18}O_n = (\delta^{18}O - \delta^{18}O_{min})/(\delta^{18}O_{max} - \delta^{18}O_{min}), \qquad (21)$$

where $\delta^{18}O_{min}$ and $\delta^{18}O_{max}$ are the minimum and maximum of $\delta^{18}O$ in the study period, respectively. We then estimate the HEP values for every 20 years using

$$HEP(t) = (1 - \delta^{18}O_n(t)) \cdot HEP_{GS} + \delta^{18}O_n(t) \cdot HEP_{GI} \qquad (22)$$

The use of $\delta^{18}O$ for the interpolation is justified because annual mean temperature is the dominant predictor for the HEP. But the largest advantage of this approach is its numerical simplicity, as otherwise GCM simulations over the entire 25,000-year period would be required, which is extremely costly. We assumed that the humans of the AUR originated from the Levant. Thus, the simulation starts with a small population Gaussian distributed in the area between the Levant and Dead Sea. While the archaeological findings from the Skhul and Qafzeh caves indicate that early AMHs already appeared here around 100 ka[15,64], the settlement of AMHs in Europe is most likely linked to the main expansion originated from this region around 45 ka[65]. The study domain covers the area of (15°W–49°E, 20°N–60°N) with a resolution of 0.5°. The model parameters used are listed in Table 2. The model outputs are saved every 10 years of the simulation.

## Uncertainties

Presented here and in the Supplementary Files is a comprehensive human dispersal model involving several factors. A detailed certainty analysis is being carried out, and several comments on the uncertainty sources can already be made. First, inaccuracies in paleoclimate model simulations may lead to uncertainties in the HEP estimates. Assessing the performance of GCMs for paleoclimate simulations is beyond the scope of this study, but with the continuing improvement of GCMs, paleoclimate simulations are expected to improve. Second, uncertainties in the HEP estimates may also arise from the incompleteness and inaccuracies of the archaeological site database. This uncertainty source has been taken into consideration in the logistic fitting in that the HEP model is trained 1000 times with 80% of randomly selected archaeological sites. This procedure allows the estimation of the HEP standard deviation which serves as a measure of this uncertainty source. Third, an important external model parameter is the maximum carrying capacity for a given culture, $\rho_{c,max}$, which influences available HEP estimates and total population. At this stage, $\rho_{c,max}$ is pre-specified. Fourth, parameters influencing the speed of human dispersal include the scaling coefficient $\alpha$ (Eq. (10)) and diffusivity $D$ (Eq. (11)). These parameters are pre-specified based on paleoanthropological research, and there is scope for further improvements. To estimate the model uncertainties, we perturbed the key

model parameters ($\rho_{c,max}$, $\alpha$ and $D$) within a reasonable range and generated an ensemble of 125 members. The ensemble mean and variance allow for example an estimate of the model uncertainty of the first arrival time, as shown in Table 1.

## Data availability

The archaeological site data[52] used in this study are available from the CRC 806 database https://crc806db.uni-koeln.de/dataset/show/crc806e1aursitesdatabase202103311617186303/. The climate reconstruction data, the Human Existence Potential, and the Our Way Model results[68] used or generated in this study have been deposited in the figshare database https://doi.org/10.6084/m9.figshare.26174980. The data behind all figures can be found in this entry as well.

## Code availability

The code of the human dispersal model and for all figures of the results[69] have been deposited in the figshare database https://doi.org/10.6084/m9.figshare.26203691.

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

## Acknowledgements

This study is funded by the Deutsche Forschungsgemeinschaft (DFG, German Research Foundation) via the Collaborative Research Centre 806 (CRC 806, Project ID 57444011) and the Ministry for Culture and Science of North Rhine-Westphalia of Germany (Profile Building 2022 PB22-081). All computations are done at the German Climate Computing Centre (DKRZ) within Project 965. We acknowledge the contribution of Dr Masoud Rostami to carrying out the paleoclimate model simulation. All plots are made with the Python packages Matplotlib[66] and Cartopy[67]. State borders are made with Natural Earth (Free vector and raster map data at naturalearthdata.com).

## Author contributions

Y. Shao and C. Wegener conceptualised the study; C. Wegener, K. Klein and Y. Shao developed the HEP model; C. Wegener developed the OWM code and carried out the simulations under the supervision of Y. Shao and G.-C. Weniger; I. Schmidt and G.-C. Weniger provided archaeological site data and contributed to the interpretations of the results; Y. Shao drafted the paper; C. Wegener prepared the graphs. All co-authors contributed to the improvement of the manuscript.

## Funding

## Competing interests

The authors declare no competing interests.
