## [Peer Review File · Nature Communications]

Reconstruction of Human Dispersal during Aurignacian on Pan-European ScaleReviewers' Comments:

Reviewer #1:

Remarks to the Author:

The authors present a computer simulation model for the spread of the Aurignacian, argued to be the first techno-complex brought into western Eurasia by modern humans. The Protoaurignacian and Early Aurignacian phases are lumped together as Phase 1 while the later Aurignacian phases are considered Phase 2 in the dispersal process. The authors established human existence potential (HEP) to account for climate/environment and its accessibility and availability for humans. HEP was developed for earlier publications by the authors to simulate Last Glacial Maximum and Aurignacian climate and human settlement patterns in Europe. The process modeled here starts with the interval of 45-43 ka cal BP and culminates ~37 ka cal BP. The model produces some interesting heuristic results regarding the spread of the Aurignacian techno-complex and expansion/contraction of associated human populations in rhythm with fluctuations in climate recorded in the Greenland ice cores.

First, I agree wholeheartedly with the authors' statement that "human dispersal is a highly convoluted process of advancement, retreat, abandonment and resettlement on different temporal and spatial scales" (lines 72-74).

For the appearance of Aurignacian sites across western Eurasia, the model results seem to show agreement with human existence potential and the known distribution of sites. But what of the blank areas on the margins of the classic core areas? Do these represent areas where human existence was not possible or where human existence has not yet been documented? The point is that site distributions in Fig. 7 cover ideal and marginal areas of HEP and maybe something else can account for that such as research bias or preservation bias. I found Fig. 7 a bit confusing. All sites appear on both a) interstadial and b) stadial condition maps. Is that intended to show that no differences in settlement patterns exist between the climate conditions? The associated text only suggests that conditions with HEP of 0.4 remain possible for human existence but it appears that while better conditions occur in refugia, people don't appear to settle there. Are they remaining in place as implied by the occupation of multicomponent "core" sites? Perhaps this could be clarified.

The problem of modern human dispersal into western Eurasia is in a continuous state of flux as new discoveries and dating has upending many previous models. The authors explicitly state that the Aurignacian represents the initial modern human techno-complex in Europe. This is no longer tenable given the current state of knowledge. The first techno-complex associated with certainty is now the Neronian with the recent findings reported from Grotte Mandrin in the Rhône Valley of France. Why is this ignored? The inclusion of Bacho Kiro, dated ~45-43 ka cal BP, as an Aur (P1) site is also questionable. The earliest modern human layers are classified as IUP, not Aurignacian. The IUP does not have the same agreed upon coherence as an industry as the Aurignacian but might loosely include a range of techno-complexes such as the Bohunician, Uluzzian, LRJ, and even Chatelperronian. The hominin associations are not as clear yet, but with the dating and stratigraphic position above the Mousterian and technological similarities with Near Eastern sites, it is likely that most if not all of these were made by modern humans. Even if there was some acculturation (e.g., the Chatelperronian) these pre-Aurignacian incursions of modern humans into western Eurasia represents the beginning of the dispersal process. The Aurignacian now represents a subsequent in-migration by *Homo sapiens* into western Eurasia. Thus, the OWM only covers a segment of the problem. How would the results differ if model were configured as a three-part one that took into account these earlier modern human dispersals?

One aspect that is left out is the potential interaction with the indigenous inhabitants, the Neanderthals. After all, the Higham et al. (Nature, vol. 512, pp.306-309, 2014) results suggest that the Mousterian continued through the Aur (P1) time interval. Hajdinjak et al. (Nature, vol. 592, pp.253-257, 2021) showed that mixing between Neanderthals and modern humans took place ~45-43 ka cal BP, prior to the appearance and spread of the Aurignacian. The model appears to assume that

modern human dispersal was unimpeded by existing human groups to simulate a purely climate and environmental process for dispersal. The rapid spread of Aur (P1) suggests no one was there to slow the advance. Or do the authors believe that the duration of Aur (P1) was sufficiently slow to account for interaction, assimilation, or extermination? Is the simulated delay in settling the southern part of Iberia due to climate or the presence of Neanderthals? What of Bajondillo and the dates for an early arrival of modern humans there? Regarding the Lapa do Picareiro site, there is a more recent exchange concerning the evidence that the authors do not cite.

Other edits:

Why on Fig. 2 are some dates given as ranges and others +/-? Shouldn't these be standardized?

Check spelling on line 312 for Kent's Cavern.

Reviewer #2:

Remarks to the Author:

Thank you for giving me the possibility to review this manuscript and for extending the deadline.

In the "Reconstruction of Human Dispersal during Aurignacian on Pan-European Scale" the authors tested a new human-dispersal model, the Our Way Model (OWM) which combines habitat suitability, human technological and cultural capacity to harness the habitat resources accounting for their availability. The output of OWS were used to reconstruct human dispersal and chronological dynamics of early Aurignacian culture. I very much appreciated that the authors tried to go beyond a simplistic climatic-driven human dispersal relationship incorporating the probability of resource exploitation and population density inside the model. However, I have some reservation about the use of anthropological record that must be addressed, in my opinion, before further consideration. The authors focused on two-time frames: 43-37 ka for the early (or Proto and Early) Aurignacian phase (Aur-P1) and 37-32 ka for the successive Aurignacian phase (Aur-P2) limiting the anthropological record to Aurignacian sites only. In the light of this, their tuning parameter procedure is linked to assumptions restricted to Aurignacian. For instance, they adopted 0.01 yr^{-1} as the reproduction rate which was estimated for Aurignacian people in Schmidt et al. (2021). Then, in the discussion Aurignacian people become representative of "human dispersal", "human density", "human population dynamics", and so on. And yet, the Aurignacian chronometrically overlaps with multiple human cultures. Douka and colleagues temporally delimit anatomically modern human Uluzzian culture between 45 and 40 ka (Douka et al. 2014). The latter time frame mostly covered the current manuscript Aur-P1 phase. Moreover, the extinction of *H. neanderthalensis* is estimated at about 40 ka even if traces of Mousterian are discovered beyond this date in Iberia Peninsula and Southern Italy (Higham et al. 2014; Ruiz et al. 2020; Zilhão et al. 2017). For the same reasons, the Levant was already occupied by different human population before the beginning of Aurignacian. Are the authors assuming that all of human dispersal is invariably valid for Aurignacian people only?

The outputs of OWM are extremely dependent on human density because some parameters such as the direction of human dispersal is hardly affected by population pressure in contiguous cells (Equation 3). How the model provides reasonable estimated of human dispersal focussing on Aurignacian sites only?

In my opinion the authors should argue about Aurignacian people dispersal not human dispersal, to say the least. But again, I'm not sure their model can account for possible competition with other human groups in term of preferred direction, diffusion, and resource availability and hence density.

The authors should provide more details about their predictor selection procedure. They started from 17 climatic variables but omitted to report the number of resulting uncorrelated variables used to calibrate their model. Moreover, I think "terrain elevation", "terrain elevation standard deviation" and "water bodies" should be considered further predictors as in the classical Species Distribution Model approaches because they could be correlated among each other or with other predictors. In this case,

the authors cannot use them as independent multipliers to adjust the estimated parameter without inflating the model predictions towards overfitting.

The authors used a classical bootstrap cross-validation (80/20 %) approach and replicate this step 1000 times. However, they did not inform the readers about the validation metric used to assess the model performance and the OWM goodness of fit. Furthermore, the input data have a strong spatial signal and are probably both temporally and spatially autocorrelated. I'm not sure their bootstrap approach is the best solution for in this case. I believe spatial or temporal block validation could be more appropriate.

I do not understand why the authors projected their model over 50-25 ka time frame if the Aurignacian started at 45 ka and finished at 32 ka as described in the paper. Could the authors clarify this doubt?

In the Equation 1, the parameter s is defined as area. Please could the authors specify what they mean as area? Cell area? In this case is it taken as constant?

Minor comment:

Authors used calibrated dates or BP dates? They did not specify it.

References:

Schmidt, I., Hilpert, J., Kretschmer, I., Peters, R., Broich, M., Schiesberg, S., ... & Maier, A. (2021). Approaching prehistoric demography: proxies, scales and scope of the Cologne Protocol in European contexts. *Philosophical Transactions of the Royal Society B*, 376(1816), 20190714.

Douka, K., Higham, T. F., Wood, R., Boscato, P., Gambassini, P., Karkanas, P., ... & Ronchitelli, A. M. (2014). On the chronology of the Uluzzian. *Journal of human evolution*, 68, 1-13.

Higham, T., Douka, K., Wood, R., Ramsey, C. B., Brock, F., Basell, L., ... & Jacobi, R. (2014). The timing and spatiotemporal patterning of Neanderthal disappearance. *Nature*, 512(7514), 306-309.

Ruiz, M. N., Jorda Pardo, J. F., Burow, C., Kehl, M., Pastoors, A., Weniger, G. C., & Wood, R. (2020). Last Neanderthal occupations at Central Iberia: the lithic industry of Jarama VI rock shelter (Valdesotos, Guadalajara, Spain). *Archaeological and Anthropological Sciences*, 12, 1-23.

Zilhão, J., Anesin, D., Aubry, T., Badal, E., Cabanes, D., Kehl, M., ... & Zapata, J. (2017). Precise dating of the Middle-to-Upper Paleolithic transition in Murcia (Spain) supports late Neanderthal persistence in Iberia. *Heliyon*, 3(11), e00435.

Reviewer #3:

Remarks to the Author:

This manuscript presents an analysis of the climatically-mediated dispersal of anatomically modern humans through Europe approximately 30-40 thousand years BP. The paper builds on previous research by the authors, developing maps of potential human carrying capacity based on climate and terrain factors, by presenting a novel model for human dispersal, which is compared to the archaeological record of the period.

This is a thorough, well-written analysis that will be of interest to archaeologists, climate scientists, and paleoecologists among other disciplines. I am not an expert in the archaeology of this region and time period, so my comments focus primarily on the quality of the climate and dispersal modeling components.

Major Comments

The authors provide a detailed description of the model, which is helpful in understanding the nuances of the study. However, there are some sections of the model description that could use further detail. Additionally, the process of model validation needs much more clarity and justification.

One suggestion for improvement is to provide one or more “null” or “baseline” models to compare to the performance of the full dispersal model. The current model includes many complex inputs and submodules — e.g. climate, terrain, accessibility, population growth, dispersal — each with uncertain mechanisms and parameters. Comparing the performance of the full model to simpler configurations would help the reader understand how the full model works and assess the added value of each additional level of complexity. For example, how well does the full model perform relative to a “naive” dispersal model that’s simply a function of geographic distance from the origin point, or that expands along a fixed number of cells each time step following the climatic gradients? Or a version of the simulation that uses present day climate fields or topography? The authors briefly mention doing something like this when comparing models of dispersal at different spatial scales, but a clearer assessment of how some or all of these various subcomponents increase the model fit to the data (or not) is crucial here. Note that this would be distinct from the ensemble approach to assessing model uncertainty that’s already used here, although that is a step in the right direction.

It would also be helpful to expand and clarify the model validation framework. It was difficult to follow when the results were referring to a single simulation run or the full ensemble spread, and in the latter case whether the ensemble results are reported for all sites or just those held out from each ensemble member. Why not use a more traditional cross validation scheme here, and report the model fit to the out-of-sample sites across each cross validation fold? A scatterplot showing predicted vs actual arrival times (using the held out samples from repeated k-fold cross validation) would be useful here to complement the numerical results presented in the table 1 and page 7 of the text. Likewise, the authors should consider using some kind of spatial cross validation structure (such as spatial-block cross validation) rather than an a-spatial resampling scheme, as the latter will yield over-optimistic estimates of model performance if there is spatial autocorrelation in the site distributions. Finally, the authors should provide a thorough discussion of the potential circularity introduced by using the archaeological site locations to both fit the human potential model and to assess the performance of the dispersal model. This is an issue that can be ameliorated through careful cross validation of the entire modeling workflow (HEP and HDM together rather than each in isolation), but it is unclear how exactly that is handled here.

Minor Comments

Code/data availability: please provide a link to the simulation and analysis code and all available input and output datasets in the revised version of this manuscript.

Lines 87-91 “Such data . . .” This sentence is a bit too dense, consider splitting into two.

Line 98 change to “integrates human- and natural-science theories and diverse data types.”

Line 100, maybe cite a paper from the 1960’s here as well to support this statement?

Line 106 change “for studying the climatic impacts” to “for studying climatic impacts”

Line 110 “Orbitally forced” instead of astronomically forced”

Line 111 “the specie distributions” to “species distributions”

Line 112, might be helpful to mention what these “special climate windows” are.

Line 113-116 this statement is vague — past models are insufficiently constrained by what? There are plenty of continental scale dispersal simulations, so what is meant by successful vs unsuccessful reconstruction in this context?

Line 149-150, what's the justification for this spatial and temporal scale? This seems to be a fairly coarse spatial resolution and fine temporal scale, all else equal. Is this for numerical stability? Ethnographic analogues? Data constraints?

Line 166, extra space after parenthesis in date.

Lines 360-364, more discussion of these high-hep-no-site areas and the potential reasons for this mismatch would be informative.

Line 420-422, is this referring to the model, data, or both? Earlier in the manuscript it was noted that the model doesn't really allow for "climate adaptation" in the traditional sense.

Line 519-520 change to "While nature offers a large amount of resources, what can be accessed is limited by the human capacity to harness it."

Line 538 missing word in "existence is also by" ?

Line 738 " R_{Δ} "

Line 8818 "human dispersal model"

Line 856 what does "homogenized to 0.5 degrees" resolution mean? From the context it seems this might refer to bilinear interpolation, but elsewhere in the required summary statements there is mention of dynamical downscaling using additional WRF. Please explain and add further detail, as the "homogenization/downscaling" process is crucial for understanding the validity of the environmental predictor layers.

Figure1, is it possible to show only sites in the corresponding time range for each panel?

Figure 4, the blue text over the flow lines are difficult to read. Consider a different font size/color, such as blue-on-white text to make the labels contrast better with the background.

Reply to Review Reports

Reviewer #1:

The authors present a computer simulation model for the spread of the Aurignacian, argued to be the first techno-complex brought into western Eurasia by modern humans. The Protoaurignacian and Early Aurignacian phases are lumped together as Phase 1 while the later Aurignacian phases are considered Phase 2 in the dispersal process. The authors established human existence potential (HEP) to account for climate/environment and its accessibility and availability for humans. HEP was developed for earlier publications by the authors to simulate Last Glacial Maximum and Aurignacian climate and human settlement patterns in Europe. The process modelled here starts with the interval of 45-43 ka cal BP and culminates ~37 ka cal BP. The model produces some interesting heuristic results regarding the spread of the Aurignacian techno-complex and expansion/contraction of associated human populations in rhythm with fluctuations in climate recorded in the Greenland ice cores.

First, I agree wholeheartedly with the authors' statement that "human dispersal is a highly convoluted process of advancement, retreat, abandonment and resettlement on different temporal and spatial scales" (lines 72-74).

For the appearance of Aurignacian sites across western Eurasia, the model results seem to show agreement with human existence potential and the known distribution of sites. But what of the blank areas on the margins of the classic core areas? Do these represent areas where human existence was not possible or where human existence has not yet been documented? The point is that site distributions in Fig. 7 cover ideal and marginal areas of HEP and maybe something else can account for that such as research bias or preservation bias.

I found Fig. 7 a bit confusing. All sites appear on both a) interstadial and b) stadial condition maps. Is that intended to show that no differences in settlement patterns exist between the climate conditions? The associated text only suggests that conditions with HEP of 0.4 remain possible for human existence but it appears that while better conditions occur in refugia, people don't appear to settle there. Are they remaining in place as implied by the occupation of multicomponent "core" sites? Perhaps this could be clarified.

Reply: We wish to thank the reviewer for the insightful question. Please note that the general agreement between the patterns of the model-predicted population density, human existence potential and archeologic site is a necessary outcome determined by the nature of the OWM framework which combines machine learning with dynamic modelling. The archaeological observations serve as a constraint on the model behavior. For example, the model should not predict high population density in areas where human existence potential (HEP) is low according to archaeological data, e.g., a desert area or dense forests. This is indeed one of the main differences between the OWM and some other models of its kind.

The OWM has two basic components, the HEP model and the Human Dispersal Model (or the HDM). The former derives the HEP using climate/environment data and the presence/absence records of archeological sites. It is a regression of the human existence probability (or potential) using climate and environment variables as predictors. The regression is a spatial-temporal interpretation of the presence/absence records of archeological sites based on the range of adaptation of a given culture to climate and environmental conditions. As a result of the regression, there are "blank areas on the margins of the classic core areas". In Shao et al. (2021), such areas are referred to as HPNS (High-Potential-No-Site) areas, while in archaeology, they are often simply called "empty" areas. As

discussed in Shao et al. (2021), we distinguish two types of HPNS areas: (a) they are non-settlement areas despite of the high HEP; and (b) they are settlement areas, but due to data bias, archeological sites have not yet been found or are not preserved. If a HPNS area is not due to research bias, then it can be viewed as a potential expansion area. In this case, the absence of sites needs to be explained based on factors other than climate and environmental conditions, e.g., humans may have not explored the area. For example, large areas with high HEP were not explored in the early part of the Aur-P1 and were settled only in the late part of the Aur-P1 and in Aur-P2.

Further considerations regarding HEP are done in the publications of Klein et al. (2023, *Assessing Climatic Impact on Transition from Neanderthal to Anatomically Modern Human Population on Iberian Peninsula: a Macroscopic Perspective*, *Sci Bull.*, <https://doi.org/10.1016/j.scib.2023.04.025>).

The dating of the archaeological sites come with large uncertainties, making it impossible to decide whether they were occupied during stadial or interstadial conditions. In our analysis for the HEP calculation, we assumed that all sites existed in interstadial times. As shown in Shao et al. (2021), this assumption is the most plausible, while the assumption that all sites existed in stadial times leading to results contradicting the common understanding (e.g. humans expanded northward during stadial times). Figure 7 shows the HEP for the prototypical extreme interstadial and stadial conditions. It is seen that interstadial times were better suitable for human existence than in stadial times. For other time slices, we estimate HEP by interpolating the HEP values for the prototypical interstadial and stadial times according to the NGRIP ice core $\delta^{18}\text{O}$ values, which serve as a proxy to temperature.

Please note that Fig. 7 only shows the HEP for prototypical stadial and interstadial conditions, not for all times. Whether humans indeed existed in high HEP areas or otherwise is then predicted using the HDM of the OWM, explicitly in terms of population density.

Changes and clarifications have been to the effect described above in the revised text .

The problem of modern human dispersal into western Eurasia is in a continuous state of flux as new discoveries and dating has upending many previous models. The authors explicitly state that the Aurignacian represents the initial modern human techno-complex in Europe. This is no longer tenable given the current state of knowledge. The first techno-complex associated with certainty is now the Neronian with the recent findings reported from Grotte Mandrin in the Rhône Valley of France. Why is this ignored? The inclusion of Bacho Kiro, dated ~45-43 ka cal BP, as an Aur (P1) site is also questionable. The earliest modern human layers are classified as IUP, not Aurignacian. The IUP does not have the same agreed upon coherence as an industry as the Aurignacian but might loosely include a range of techno-complexes such as the Bohunician, Uluzzian, LRJ, and even Chatelperronian. The hominin associations are not as clear yet, but with the dating and stratigraphic position above the Mousterian and technological similarities with Near Eastern sites, it is likely that most if not all of these were made by modern humans. Even if there was some acculturation (e.g., the Chatelperronian) these pre-Aurignacian incursions of modern humans into western Eurasia represents the beginning of the dispersal process. The Aurignacian now represents a subsequent in-migration by *Homo sapiens* into western Eurasia. Thus, the OWM only covers a segment of the problem. How would the results differ if model were configured as a three-part one that took into account these earlier modern human dispersals?

Reply: The term IUP (Initial Upper Palaeolithic) vaguely describes a very heterogeneous group of lithic inventories often small in size and excavated a long time ago, which are attributed to both AMH and Neanderthal, depending on region or research group. Its technological definition and its

duration are highly controversial as well. In the last years, paleogenetic data with significantly higher resolution than technological data of the lithic inventories gave new input to the discussion. Data from Oase 1 (Fu et al. 2015), Bacho Kiro (Hajdinjak et al. 2021) and Zlatý kůň (Prüfer et al. 2021) demonstrate at least two dispersal impulses of AMH that reached Eastern Europe, but according to the current knowledge, did not lead to a significant settlement for reasons still unknown and groups became extinct (Vallini et al. 2022). Only in Bacho Kiro was a small lithic inventory recovered, attributed to an IUP. Studies on the Châtelperronian in northern Spain also indicate a collapse of populations. Here after the Middle Palaeolithic with a subsequent recolonization by the Neanderthal groups of the Châtelperronian (Rios-Garaizar et al. 2022). The situation of the IUP in the Rhone Valley with the Neronian is even more complex. The data from the Grotte du Mandarin with an attribution of the finds to AMH (Slimak et al. 2022) are contrasted by the results from the Abri du Maras (Ruebens et al. 2022), which postulate an attribution of the Neronian to Neanderthals. In Italy, the IUP complex of the Uluzzian has been connected to AHMs due to human remains in one site only (Benazzi et al. 2011).

The situation of the IUP is thus highly complex and still unresolved. However, the archaeological records so far suggest a great dynamic of human dispersal processes with regular collapses of populations in a narrow time window of a few thousand years at the most. Since there is no confirmed interstratification of the techno-complexes in the sites, the burden of proof of the timing of these processes lies primarily on radiocarbon chronology. The dating process and its subsequent evaluation has been constantly improved, but uncertainty of dating beyond 40 ka remains an enormous challenge that is well known but tends to be pushed into the background when interpreting the prehistoric processes. Numerous statistical procedures are regularly presented to better assess the uncertainty of dating, e.g., Djakovic et al. (2022) for the latest information. However, they cannot fully compensate for the low quality of many dates and might be, in our opinion, often too optimistic.

Therefore, we intentionally limit our study here to the AUR. Its definition is also not free of dispute but much more homogeneous compared to the IUP, and therefore a solid starting point for a review of the early dispersal process of AMH in Europe. We have modified the introduction to clarify that the focus of this study is on the AUR, as the first techno-complex of AMH, which successfully spread to western and south-western Europe as a significant population turn over.

In the revised version, we have added two sensitivity experiments dedicated to addressing the question of IUP (BS7 and BS8 in Table 4). The OWM simulation suggests that a critical population density is necessary to initiate the human dispersal process and a sufficiently high reproduction rate is necessary for the dispersal to be sustainable. A population with density and reproduction below critical values is vulnerable to changes. In numerical experiment BS8, we have assumed existence of IUP humans at 50 kyr cal BP at four sites. Due to the low cultural carrying capacity and low reproduction rate (we selected), the IUP human population density remained too low to compete with the large inflow of humans of the Aurignacian at the time as they firmly settled in central Europe (around 42 kyr cal BP). If our assumptions (with respect to the cultural carrying capacity and reproduction rate) are acceptable (which we certainly think are reasonable), then the IUP techno-complexes with a low population density would be much less resilient to changes of climate and environment and could not achieve sustainable dispersal on pan-European scale. The results of numerical experiments BS7 and BS8 show that the presence of IUP humans would not have a significant impact on the dispersal of humans of the Aurignacian.

Yaping Shao, Heiko Limberg, Konstantin Klein, Christian Wegener, Isabell Schmidt, Gerd-Christian Weniger, Andreas Hense, Masoud Rostami, Human-existence probability of the Aurignacian techno-complex under extreme climate conditions, *Quaternary Science Reviews*, Volume 263, 2021, 106995, ISSN 0277-3791, <https://doi.org/10.1016/j.quascirev.2021.106995>.

Fu, Q., Posth, C., Hajdinjak, M. et al. The genetic history of Ice Age Europe. *Nature* **534**, 200–205 (2016). <https://doi.org/10.1038/nature17993>

Hajdinjak, M., Mafessoni, F., Skov, L. et al. Initial Upper Palaeolithic humans in Europe had recent Neanderthal ancestry. *Nature* **592**, 253–257 (2021). <https://doi.org/10.1038/s41586-021-03335-3>

Prüfer, K., Posth, C., Yu, H. et al. A genome sequence from a modern human skull over 45,000 years old from Zlatý kůň in Czechia. *Nat Ecol Evol* **5**, 820–825 (2021). <https://doi.org/10.1038/s41559-021-01443-x>

Vallini L, Marciani G, Aneli S, et al (2022) Genetics and Material Culture Support Repeated Expansions into Paleolithic Eurasia from a Population Hub Out of Africa. *Genome Biology and Evolution* 14(4). <https://doi.org/10.1093/gbe/evac045>, URL <https://doi.org/10.1093/gbe/evac045>, evac045

One aspect that is left out is the potential interaction with the indigenous inhabitants, the Neanderthals. After all, the Higham et al. (*Nature*, vol. 512, pp.306-309, 2014) results suggest that the Mousterian continued through the Aur (P1) time interval. Hajdinjak et al. (*Nature*, vol. 592, pp.253-257, 2021) showed that mixing between Neanderthals and modern humans took place ~45-43 ka cal BP, prior to the appearance and spread of the Aurignacian. The model appears to assume that modern human dispersal was unimpeded by existing human groups to simulate a purely climate and environmental process for dispersal. The rapid spread of Aur (P1) suggests no one was there to slow the advance. Or do the authors believe that the duration of Aur (P1) was sufficiently slow to account for interaction, assimilation, or extermination? Is the simulated delay in settling the southern part of Iberia due to climate or the presence of Neanderthals? What of Bajondillo and the dates for an early arrival of modern humans there? Regarding the Lapa do Picareiro site, there is a more recent exchange concerning the evidence that the authors do not cite.

Reply: Our reply to the query is as follows. We are dealing with two aspects here, one archaeological and the other technical.

Our studies on the Iberian Peninsula, for example (Klein et al. 2023), show that the end of the Middle Palaeolithic is indeed accompanied by a decline in Neanderthal populations or even a settlement gap. In northern Italy (Riel-Salvatore et al. 2022) and in Central Europe, numerous stratigraphy also indicate a similar phenomenon. These observations reduce the likelihood of Neanderthal interaction with AMH the further west we go in Europe. The estimate of the number of generations since the last common admixture event with Neanderthals is about 6-8 generations for the oldest individual from the IUP in Bacho Kiro and in Oase (Hajdinjak et al. 2021). In the slightly older find from Zlaty Kun, the estimate is about 70 generations. This puts us in a time window of less than 200 to less than 2,000 years and brings us to the limits of the resolving power of palaeogenetics. But once again, it is clear that we are dealing with population changes that are as short-term as the rapid climate changes that are running behind them.

We have discussed Cueva de Bajondillo and Lapa do Picareiro in detail in Klein et al. (2023). The early dating of Bajondillo may rightly be doubted which is also considered by other researchers. For Lapa do Picareiro we did special test of the HEP model and found that the attribution to an AUR P1 has low probability only. We explain this in the text now.

The current version of OWM does not simulate the interactions of several groups of techno complexes or even other types of homini, as we would require more data (e.g. for HEP of other

homini types) and have to make more assumptions regarding their interactions (coexistence, assimilation, extermination). Without that, even though a mathematical formulation for group interactions can be made, a realistic simulation using the model would be difficult. More development work is now on the way by this research group.

The delay of settlement in the southern part of Iberia is forced by the HEP. The climate conditions in Iberia were different compared to the central regions of Europe which features most of the sites. The logarithmic regression of the HEP evaluates the lack of sites and the different climate in this region as not suitable for the early Aurignacian. The later Aurignacian site distribution is more spread out, indicating a different adaptation that allows for the spread into the southern Iberian along the Mediterranean coast. A detailed analysis of Klein et al. (2023) shows that it is unlikely that Neanderthals and AMHs extensively coexisted on the Iberian Peninsula and hence it is also highly unlikely that the likely presence of Neanderthals significantly affected the expansion of the AMHs there. The slow expansion of the AMHs into Iberia is more probably due to the constellation of human expansion and climate change. At the time of GI10, the AMHs just arrived in northern Spain via southern France but were mostly settled in the Cantabrian coastal areas. It is highly likely that the AMH population did not expand southward to cross the Ebro valley at the time, which represents the northern boundary of a large territory on the Iberian Peninsula south of the Ebro that was hostile for the AMH settlement during Aur-P1. The AMHs arrived northern Iberia was soon faced by the prolonged cold phase of GS9/HE4, which lasted almost 3000 years, and the AMH population on the pan-European scale was retreating (as shown in Fig. 4), and the conditions for the existence of the AMHs in Iberia were further deteriorated and the suitable areas for their existence reduced. Thus, during the entire time period from GI10 to GS10-9/HE4, there was little chance for the southward expansion of the AMHs. Only after GS9/HE4, in the second phase of the AUR, the AMHs headed southward along the coast in today's Valencia and Andalucía, while the interior of Iberian remained to be unsettled.

We have made revisions to the effect above in (1) the discussions of the results in the main text and (2) the discussions of model uncertainties in the supplementary material

Other edits:

Why on Fig. 2 are some dates given as ranges and others +/-? Shouldn't these be standardized?

Reply: Thanks. This is because that the site dating was used as presented in their respective original publications, in some of which a range is given, while in others in +/- . Unfortunately, we were not able to provide a standard error margin.

Check spelling on line 312 for Kent's Cavern.

Reply: Thanks. We have corrected the spelling.

Reviewer #2:

The authors focused on two-time frames: 43-37 ka for the early (or Proto and Early) Aurignacian phase (Aur-P1) and 37-32 ka for the successive Aurignacian phase (Aur-P2) limiting the anthropological record to Aurignacian sites only. In the light of this, their tuning parameter procedure is linked to assumptions restricted to Aurignacian. For instance, they adopted 0.01 yr^{-1} as the reproduction rate which was estimated for Aurignacian people in Schmidt et al. (2021). Then, in the discussion Aurignacian people become representative of “human dispersal”, “human density”, “human population dynamics”, and so on. And yet, the Aurignacian chronometrically overlaps with multiple human cultures. Douka and colleagues temporally delimit anatomically modern human Uluzzian culture between 45 and 40 ka (Douka et al. 2014). The latter time frame mostly covered the current manuscript Aur-P1 phase. Moreover, the extinction of *H. neanderthalensis* is estimated at about 40 ka even if traces of Mousterian are discovered beyond this date in Iberia Peninsula and Southern Italy (Higham et al. 2014; Ruiz et al. 2020; Zilhão et al. 2017). For the same reasons, the Levant was already occupied by different human population before the beginning of Aurignacian. Are the authors assuming that all of human dispersal is invariably valid for Aurignacian people only?

Reply: We thank the referee for this comment. First, we agree with the referee that the use of expressions of “human dispersal”, “human density” and “human population dynamics” is not entirely accurate. They should be understood as “dispersal of humans of the Aurignacian” etc. As these are very long phrases, we add in the beginning of the text that “human dispersal” in our study refers to the dispersal of humans of the Aurignacian unless otherwise stated. It should also be stated that the parameters and indeed the human existence potential (HEP) used in this study are specific for the Aurignacian culture. A different set of parameters and HEP need to be used for a different techno-complex. We have made changes in the text for clarification.

We have now discussed in the text the highly complex situation of the MP, IUP and AUR which remains partly unsolved. And we have tried to make it clear why we have deliberately limited our effort to the AUR as the first techno-complex of AMH which successfully spread to western and south-western Europe as a significant population turn over. We have now discussed the data from Oase 1 (Fu et al. 2015), Bacho Kiro (Hajdinjak et al. 2021) and Zlatý kůň (Prüfer et al. 2021), that demonstrate at least two dispersal impulses of AMH that reached Eastern Europe prior to the AUR. But according to current knowledge did not lead to a significant settlement for reasons still unknown and groups got extinct (Vallini et al. 2022). As mentioned in our reply to Review 1, we have carried out additional numerical experiments BS7 and BS8 to address the issue whether IUP humans could have significantly affected the dispersal of humans of the Aurignacian.

We have discussed Cueva de Bajondillo and Lapa do Picareiro in detail in Klein et al. (2023). The early dating of Bajondillo may rightly be doubted which is also considered by other authors. For Lapa do Picareiro, we did specific test of the HEP model and found that the attribution to an AUR P1 has low probability only. We have now explained this in the text.

The outputs of OWM are extremely dependent on human density because some parameters such as the direction of human dispersal is hardly affected by population pressure in contiguous cells (Equation 3). How the model provides reasonable estimated of human dispersal focusing on Aurignacian sites only?

In my opinion the authors should argue about Aurignacian people dispersal not human dispersal, to say the least. But again, I’m not sure their model can account for possible competition with other human groups in term of preferred direction, diffusion, and resource availability and hence density.

Reply: As stated in our reply to Review I, the current version of the OWM does not simulate the interactions of several groups of techno-complexes or other types of homini, as we would require more data and have to make many more assumptions regarding their interactions (coexistence, assimilation, extermination). Without that, credible results are difficult to achieve. The Aurignacian techno-complex is used here as the archaeological data coverage was large enough to allow a reasonable estimation of the HEP which is required as an input for the human dispersal model. One alternative is to use NPP as a driver for the human dispersal model, as suggested in some other studies, but we know this method has profound pitfalls. For example, because NPP is independent of culture, if we use the NPP as a driver to model the dispersal of Neanderthals and AMHs, we will end up with the same distribution pattern for both populations.

In the OWM, human dispersal is affected by population pressure gradient and this is described with Equations 3 and 9. This is reflected both in the drift and diffusion processes. For example, the drift velocity is proportional to the gradient of available human existence potential, which is a function of population density. It is in available HEP, namely, the variable denoted with Φ_{Av} , that the population pressure is imbedded, as shown in Fig. 6. After reading the text again, we feel that the description on this issue appears to be sufficient clear and we have therefore added no further text in the revised version.

The authors should provide more details about their predictor selection procedure. They started from 17 climatic variables but omitted to report the number of resulting uncorrelated variables used to calibrate their model. Moreover, I think “terrain elevation”, “terrain elevation standard deviation” and “water bodies” should be considered further predictors as in the classical Species Distribution Model approaches because they could be correlated among each other or with other predictors. In this case, the authors cannot use them as independent multipliers to adjust the estimated parameter without inflating the model predictions towards overfitting.

Reply: The techniques for HEP estimates have been presented in detail in the earlier papers by the same group (Klein et al., 2021, Shao et al. 2021b, Klein et al. 2023). Care has been taken to avoid collinearity of the predictors. Particularly relevant for this study is the work of Shao et al. 2021b, where the details of HEP estimates for the Aurignacian culture are given. After lengthy analysis, four uncorrelated (or weakly correlated) variables are used for HEP estimates, as mentioned in the text, they are T_m (annual mean temperature), T_δ (temperature seasonality), R_m (annual rainfall) and R_δ (rainfall seasonality).

Certain environmental parameters, such as water bodies, mountain terrains, presence of dense forests, which affect human existence, are not used as HEP predictors, but are considered separately as modification factors. For example, climate conditions over a large water body may be perfectly suited for human existence, but the area may be not accessible for certain techno-complexes. Therefore, for large water bodies, HEP is reduced to zero by applying a modification function. Similarly, terrain elevation and complexity are not used as HEP predictors, but considered in computing the accessible HEP. These environment factors are not used as HEP predictors, as they are more or less stationary on the time scale of cultural evolution, and their influences on human existence are culture dependent, e.g., areas not accessible for more primitive technologies may become accessible for more sophisticated techno-complexes. At this stage of our model development, the representation of different technologies and their impact is still rudimentary and should be further improved. We added some explanations to the section where the calculation of HEP is described.

The authors used a classical bootstrap cross-validation (80/20%) approach and replicate this step 1000 times. However, they did not inform the readers about the validation metric used to assess the model performance and the OWM goodness of fit. Furthermore, the input data have a strong spatial signal and are probably both temporally and spatially autocorrelated. I'm not sure their bootstrap approach is the best solution for in this case. I believe spatial or temporal block validation could be more appropriate.

Reply: We would like to mention that the OWM has two components, one is the HEP model (a machine-learning procedure for determining the spatial-temporal variable of HEP) and the dynamic human dispersion model (HDM). The HEP model employs essentially a bootstrap cross-validation. Again, the details to the estimation of the HEP have been presented in the previous studies of Klein et al. (2021), Shao et al. (2021b) and Klein et al. (2023). As the focus of this study is on the reconstruction of the dispersal of humans of the Aurignacian, we have to limit the length of this paper without too much repeating of the earlier studies. However, we agree with the Referee that we should provide sufficient information also here. We have now added a section for the validation of the HEP estimates relevant for the Aurignacian.

To avoid the spatial over-fitting due to stacked sites, a down sampling procedure is implemented for training the HEP model, namely, we only use the "presence/absence" records of the archaeological sites on the grid of the climate data (with resolution of 0.5° lat/lon), instead of the archaeological sites directly. For example, if a grid cell has several archaeological sites, the grid cell is considered only as one presence record.

We have also thought about the "spatial block validation method" (e.g., Roberts et al. 2017: Cross-validation strategies for data with temporal, spatial, hierarchical, or phylogenetic structure. *Ecography* 40, 913–929), but this method may not be suitable for our study, as it leads to large variations in the HEP patterns due to the fact that the neither the "presence" records (the number of which is small) no the climate data are spatially homogeneously distributed. Considering that HEP represents the adaptation of humans of a given techno-complex to climate and environment, the spatial block validation method does not seem to be suitable for our purpose here.

I do not understand why the authors projected their model over 50-25 ka time frame if the Aurignacian started at 45 ka and finished at 32 ka as described in the paper. Could the authors clarify this doubt?

Reply: this is a typo and is now corrected accordingly.

In the Equation 1, the parameter s is defined as area. Please could the authors specify what they mean as area? Cell area? In this case is it taken as constant?

Reply: The variable $s_{i,j}$ denotes the cell area of the grid point (i, j). As the grid points are spatially fixed, the cell areas are constant in time but slightly different in space for different cells due to the distortion of the latitude/longitude grid. Using the human dispersal model, the human population density ρ (in humans per unit area) is calculated. It follows that the total number of humans per grid cell is $\rho_{i,j} \times s_{i,j}$. We have changed the text to make this clearer.

Minor comment:

Authors used calibrated dates or BP dates? They did not specify it.

Reply: Thanks. They are cal BP dates. We have now modified the text.

References:

- Schmidt, I., Hilpert, J., Kretschmer, I., Peters, R., Broich, M., Schiesberg, S., ... & Maier, A. (2021). Approaching prehistoric demography: proxies, scales and scope of the Cologne Protocol in European contexts. *Philosophical Transactions of the Royal Society B*, 376(1816), 20190714.
- Douka, K., Higham, T. F., Wood, R., Boscato, P., Gambassini, P., Karkanas, P., ... & Ronchitelli, A. M. (2014). On the chronology of the Uluzzian. *Journal of human evolution*, 68, 1-13.
- Higham, T., Douka, K., Wood, R., Ramsey, C. B., Brock, F., Basell, L., ... & Jacobi, R. (2014). The timing and spatiotemporal patterning of Neanderthal disappearance. *Nature*, 512(7514), 306-309.
- Ruiz, M. N., Jorda Pardo, J. F., Burow, C., Kehl, M., Pastoors, A., Weniger, G. C., & Wood, R. (2020). Last Neanderthal occupations at Central Iberia: the lithic industry of Jarama VI rock shelter (Valdesotos, Guadalajara, Spain). *Archaeological and Anthropological Sciences*, 12, 1-23.
- Zilhão, J., Anesin, D., Aubry, T., Badal, E., Cabanes, D., Kehl, M., ... & Zapata, J. (2017). Precise dating of the Middle-to-Upper Paleolithic transition in Murcia (Spain) supports late Neandertal persistence in Iberia. *Heliyon*, 3(11), e00435.

Reviewer #3:

Major Comments

The authors provide a detailed description of the model, which is helpful in understanding the nuances of the study. However, there are some sections of the model description that could use further detail. Additionally, the process of model validation needs much more clarity and justification.

Reply: We greatly appreciate the insightful suggestions of the referee. In response to the suggestions of the referee, we have reorganized Section 3.5 (by adding three new subsections) which now reads like

Section 3.5 Model Uncertainty and Ensemble Simulation

3.5.1 Idealized Experiments

3.5.2 IUP Experiment

3.5.3 Validation Strategy

3.5.4 Other Uncertainties

One suggestion for improvement is to provide one or more “null” or “baseline” models to compare to the performance of the full dispersal model. The current model includes many complex inputs and submodules — e.g. climate, terrain, accessibility, population growth, dispersal — each with uncertain mechanisms and parameters. Comparing the performance of the full model to simpler configurations would help the reader understand how the full model works and assess the added value of each additional level of complexity. For example, how well does the full model perform relative to a “naive” dispersal model that’s simply a function of geographic distance from the origin point, or that expands along a fixed number of cells each time step following the climatic gradients? Or a version of the simulation that uses present day climate fields or topography? The authors briefly mention doing something like this when comparing models of dispersal at different spatial scales, but a clearer assessment of how some or all of these various subcomponents increase the model fit to the data (or not) is crucial here. Note that this would be distinct from the ensemble approach to assessing model uncertainty that’s already used here, although that is a step in the right direction.

Reply: Thanks for this suggestion. In Section 3.5.1 Idealized Experiments, we presented the results for idealized experiments. Indeed, the “baseline simulations” and the performance of the Human Dispersion Model, a component of the OWM, have been discussed in great detail in Wegener (2021), now cited in the revised version. Following the suggestions of Review III, we present the following “baseline simulations” in the supplementary material and made additional discussions.

Baseline simulation	Modell	Initial Field	HEP
BS1	Full	Lavent	Uniform
BS2	Full	-	N-S gradient
BS3	Full	-	E-W gradient
BS4	No drift velocity	-	HEP model derived
BS5	No diffusion	-	-
BS6	No production	-	-
BS7	Full, reduced birth rate	Random, low density	-
BS8	Full, reduced birth rate	Four IUP sites, low density	HEP model for AUR-P1, reduced cultural carrying capacity

It would also be helpful to expand and clarify the model validation framework. It was difficult to follow when the results were referring to a single simulation run or the full ensemble spread, and in the latter case whether the ensemble results are reported for all sites or just those held out from each ensemble member. Why not use a more traditional cross validation scheme here, and report the model fit to the out-of-sample sites across each cross-validation fold? A scatterplot showing predicted vs actual arrival times (using the held-out samples from repeated k-fold cross validation) would be useful here to complement the numerical results presented in the table 1 and page 7 of the text. Likewise, the authors should consider using some kind of spatial cross validation structure (such as spatial-block cross validation) rather than an a-spatial resampling scheme, as the latter will yield over-optimistic estimates of model performance if there is spatial autocorrelation in the site distributions. Finally, the authors should provide a thorough discussion of the potential circularity introduced by using the archaeological site locations to both fit the human potential model and to assess the performance of the dispersal model. This is an issue that can be ameliorated through careful cross validation of the entire modeling workflow (HEP and HDM together rather than each in isolation), but it is unclear how exactly that is handled here.

Reply: This point is fully taken. The data available for validation are as follows:

- (1) The location of archaeological sites for the validation of HEP estimates; and
- (2) A handful of sites with dating for the validation of OWM simulation of arrival time.

As mentioned in our reply to Review 1 and 2, the validation of the HEP model has been done in the previous publications of Klein et al. (2022), Shao et al. (2022b) and Klein et al. (2023). In the revised paper, we have added some description on this issue. Independent validation of the HDM is only possible using the arrival time, independent estimates of population density (which do not seem to exist) and expert knowledge. While the spatial correlation between simulated population density and observed population density is highly desirable, the latter data are not (to the best of our knowledge) available.

While we believe a scatter-plot is a good idea, but it would produce yet another graph in the main text and generating some repeating information to the table we have already inserted. Thus, we have not added this new graph, also for the reason that the number of data points is very small.

We have now discussed the potential circularity. Again, HEP derived from the presence/absence records of archaeological sites provide us only the information under which condition humans likely existed, the HDM simulations provide the information whether, when and how the potential is realized, in terms of population density. Obviously, HEP cannot be used to validate the population density, but independent information from archaeology (e.g. arrival time and independently estimated population density) needs to be used for model validation. We have attempted to clarify these issues in the revised text. In the revised manuscript, we added a graph showing the model-simulated population density at 41 kyr cal BP for all Aur-P1 sites, when the settlement for the first Aurignacian phase reached its maximum extend, and at 38 kyr cal BP for all Aur-P2 sites, when the settlement in the second Aurignacian phase reached its maximum extend.

Minor Comments

Reply: Minor comments are considered. Much appreciated.

Code/data availability: please provide a link to the simulation and analysis code and all available input and output datasets in the revised version of this manuscript.

Reply: in the process of the publication, we will deposit the code and data in the database maintain at crc806db.uni-koeln.de

- Lines 87-91 “Such data . . .” This sentence is a bit too dense, consider splitting into two.

Reply: Thanks. We modified the line

- Line 98 change to “integrates human- and natural-science theories and diverse data types.”

Reply: Thanks. We modified the line as suggested.

- Line 100, maybe cite a paper from the 1960’s here as well to support this statement?

Reply: Yes. We have now added the references from the 1960s.

- Line 106 change “for studying the climatic impacts” to “for studying climatic impacts”

Reply: Thanks. We modified the line as suggested.

- Line 110 “Orbitally forced” instead of astronomically forced”

Reply: The expression “astronomically forced “ was used in the original paper of Timmermann et al. (2022). We have therefore kept “astronomically forced” unchanged to avoid any possible additional interpretation of their work, although we agree that “orbital forced” appears to be what is meant in the original paper.

- Line 111 “the specie distributions” to “species distributions”

Reply: Thanks. We modified the line as suggested.

- Line 112, might be helpful to mention what these “special climate windows” are.

Reply: Thanks. We added a line to this.

- Line 113-116 this statement is vague — past models are insufficiently constrained by what?

Reply: Good point. We slightly modified the line.

- There are plenty of continental scale dispersal simulations, so what is meant by successful vs unsuccessful reconstruction in this context?

Reply: We beg to differ. We certainly are not aware of any examples similar as report here, but we have removed the word “successful”.

- Line 149-150, what’s the justification for this spatial and temporal scale? This seems to be a fairly coarse spatial resolution and fine temporal scale, all else equal. Is this for numerical stability? Ethnographic analogues? Data constraints? *Both data constraints (climate) and numerical stability (time steps).*

Reply: The main reason for this is that the numerical grid for the climate model has a resolution of about 50 km. Implicitly, this is related how we interpret an archaeological observation. We have assumed an archaeological site implies existence of humans in the surrounding areas of about 50 km x 50 km. For local and regional scale simulations, for which we have more data (e.g. Iberia), the model resolution can be increased (e.g. to 10 km). Without diverting from the line of discussion too much, we only mentioned in the revised text that the climate model has a grid resolution of 50 km and did not elaborate too much.

- Line 166, extra space after parenthesis in date.

Reply: Thanks. Corrected.

- Lines 360-364, more discussion of these high-hep-no-site areas and the potential reasons for this mismatch would be informative.

Reply: We have no added some discussions, in Section 3.3.

- Line 420-422, is this referring to the model, data, or both? Earlier in the manuscript it was noted that the model doesn't really allow for “climate adaptation” in the traditional sense.

Reply: This version of OWM does not model human adaption to climate, but the HEP data may reflect the fact that humans may be adapted to a certain climate. In Aur-P2, compared with Aur-P1, more sites exist in colder regions. As a consequence, HEP for Aur-P2 has higher values than that for Aur-P1 in colder climate. e OWM does not explicitly model Refers to the model. The logarithmic regression of the HEP simulates a “climate adaptation” to a specific set of climate data without any temporal change, thus there is no classical adaption over time.

- *Line 519-520 change to “While nature offers a large amount of resources, what can be accessed is limited by the human capacity to harness it.”*

Reply: Thanks. We modified the text as suggested.

- *Line 538 missing word in “existence is also by” ?*

Reply: Thanks. We corrected the line.

- *Line 738 “ R_{δ} ”*

Reply: Thanks. We corrected the line.

- *Line 8818 “human dispersal model”*

Reply: Thanks. We corrected the line.

- *Line 856 what does “homogenized to 0.5 degrees” resolution mean? From the context it seems this might refer to bilinear interpolation, but elsewhere in the required summary statements there is mention of dynamical downscaling using additional WRF. Please explain and add further detail, as the “homogenization/downscaling” process is crucial for understanding the validity of the environmental predictor layers.*

Reply: Thanks. We modified the text.

- *Figure1, is it possible to show only sites in the corresponding time range for each panel?*

Reply: This would be ideal, but unfortunately not, as the archaeological dating does not have the required temporal resolution.

- *Figure 4, the blue text over the flow lines are difficult to read. Consider a different font size/color, such as blue-on-white text to make the labels contrast better with the background.*

Reply: Thanks. We have tried to improve the quality of the graph.

Reviewers' Comments:

Reviewer #1:

Remarks to the Author:

Comments on revised manuscript:

The IUP experiment added to the resubmission is apparently to partially address concerns about the omission of this important phase of modern human dispersal. I'm not sure that it strengthens the manuscript because it raises additional concerns and questions. First, I wonder how the results would look if Uluzzian sites from Italy and Neronian sites from southern France were included as modern human associated IUP. Second, why should the birthrate for these modern humans be 1/10th that of Aurignacian modern humans? If there really has emerged a consensus that demographic factors explain the demise of Neanderthals and success of modern humans (Vaesen et al. 2021), why did these modern humans not have the same reproductive capacity as Aurignacian modern humans? Is it due to hybridization and possibly lowered reproductive success? The limited skeletal evidence dated to the IUP-Aur-P1 timeframe shows that these earlier modern humans had recent Neanderthal ancestry suggesting that the demographic assimilation of Neanderthal populations and their widespread replacement was taking place during this period. Neanderthals were almost entirely gone by the Aur-P2 so what we may be looking at is a replacement and assimilation of the earlier modern human lineage by a subsequent one, as explicitly implied by the authors' climate model.

Having said that I think the reorganization of section 3.5 was helpful to clarify aspects of the model.

Line 061: should be Mandrin, not Mandarin

Line 980: "Little Asia?" Is that Asia Minor? Maybe just continue to use Anatolia.

I still have issues with the authors' assumptions regarding the spread of modern humans.

- 1) Aurignacian represents dispersal/migration/colonization of empty landscapes by modern humans. Equally plausible explanation for such a unified package is the adoption of it through existing exchange networks.
- 2) IUP at Bacho Kiro known largely from old excavations but recent excavations demonstrated the age and integrity of lithic assemblages.
- 3) Grotte Mandrin represents the only place where a hominin is associated with the Neronian. At Abri du Maras, the Neanderthal association is assumed despite a lack of hominin remains, including bones, sedaDNA, and proteomes. The Mandrin finds cannot be dismissed through contrast with Maras.
- 4) The Oase 1 specimen, dated 42-37 ka calBP, once served as the best evidence to associate the Aurignacian with AMH. Fu et al. (2015) showed that it had recent Neanderthal ancestry and wasn't closely related to later Europeans. Hublin et al. (2020) suggested that it might be older because the date was produced using antiquated pretreatment protocols. Posth et al. (2023) place the oldest modern human closely related to subsequent Europeans at Kostenki ~37 ka calBP and Goyet ~35 ka calP. Thus, it is plausible that the Aurignacian itself represents multiple migrations of people, some of which were unsuccessful in the sense that they did not contribute genetic heritage to existing European lineages. The authors' Aur-P1 may be no more representative than any dismissed IUP migrations. Aur-P2 would be the first truly successful (at least in terms of impact on subsequent Europeans) modern human dispersal into Europe. This scenario is explained by the authors' demographic estimates for a major population decline between Aur-P1 and Aur-P2.

Some developments since submission that might influence the authors' thinking about the Chatelperronian:

Gicqueau, A., Schuh, A., Henrion, J., Viola, B., Partiot, C., Guillon, M., Golovanova, L., Doronichev, V., Gunz, P., Hublin, J.-J., Maureille, B., 2023. Anatomically modern human in the Châtelperronian hominin collection from the Grotte du Renne (Arcy-sur-Cure, Northeast France). *Scientific Reports* 13, 12682.

Henrion, J., Hublin, J.-J., Maureille, B., 2023. New Neanderthal remains from the Châtelperronian-attributed layer X of the Grotte du Renne (Arcy-sur-Cure, France). *Journal of Human Evolution* 181, 103402.

Plus, a third that will cause discomfort to some prehistorians:

Slimak, L., 2023. The three waves: Rethinking the structure of the first Upper Paleolithic in Western Eurasia. *PLOS ONE* 18, e0277444.

Reviewer #2:

Remarks to the Author:

Dear editor,

thank you for giving me the possibility to review this manuscript.

In this R2 of the manuscript "Reconstruction of Human Dispersal during Aurignacian on Pan-European Scale", the authors increased the discussions about the most critical points in the main text. However, I was not convinced that they gave significant improvements to the manuscript. The authors did not solve my skepticism about the articles.

First, I asked more details about the HEP models published elsewhere. The authors have done a simple copy and paste of several paragraphs. Consequently, it is very unclear what the authors have really done in their manuscript and/or what they have retrieved from the previous model calibration setup.

For instance, Shao et al. 2021 selected 4 uncorrelated predictors: BIO1, BIO4, BIO13, and BIO15 and in the current paper, I read:

"The selection the predictors requires they are not collinear and allow the logistic regression to distinguish the presence/absence records. We use the hierarchical correlation cluster technique to examine the collinearity between the candidate predictors. Shao et al(2021b) showed that Tm (annual mean temperature), T δ (temperature seasonality), Rm (annual rainfall) and R δ (rainfall seasonality) are the most suitable predictors for the Palaeolithic humans."

Although, I assume that the HEP was already calibrated and validated in Shao et al. 2021, the authors listed a different list of predictors since Rm /BIO 12 ("annual rainfall") was never selected by Shao and colleagues.

As further clarification, I would have liked to know what it means a "subset of the grid cells where no archaeological site exists as "absence cells". There is a lot of literature about the number and the appropriate locations of absence points used to calibrate a model (Barbet-Massin et al. 2012) but the authors did not give any details about it. Furthermore, I would point out to the authors that they used the human fossil records to define the empty cells. Consequently, they cannot rely on the "true" absence points but "pseudo-absence" datapoints instead (Lobo et al. 2010) because of the dataset limitations.

Barbet-Massin, M., Jiguet, F., Albert, C. H., & Thuiller, W. (2012). Selecting pseudo-absences for species distribution models: How, where and how many?. *Methods in ecology and evolution*, 3(2), 327-338.

Lobo, J. M., Jiménez-Valverde, A., & Hortal, J. (2010). The uncertain nature of absences and their importance in species distribution modelling. *Ecography*, 33(1), 103-114.

Second, I asked to verify for spatial correlation in native data by using another cross-validation scheme. The authors replied to me:

"We have also thought about the "spatial block validation method" (e.g., Roberts et al. 2017: Cross

validation strategies for data with temporal, spatial, hierarchical, or phylogenetic structure. *Ecography* 40, 913–929), but this method may not be suitable for our study, as it leads to large variations in the HEP patterns due to the fact that the neither the “presence” records (the number of which is small) no the climate data are spatially homogeneously distributed.”

The authors did not follow my suggestion and with the last sentence, they practically admitted their dataset is spatially biased.

Eventually, the OWM has two components, the HEP model (a machine-learning procedure for determining the spatial-temporal variable of HEP) and the dynamic human dispersion model (HDM) whereby the first has already calibrated in Shao et al. 2021. Hence, the “new” contribution of the authors for OWM is limited to the HDM and accessibility parameter. Anyway, I strongly suspect they created an overestimation of human density (restricted to Aurignacian only) because their OWM does not take into account the interaction with other human groups (Neanderthals and human with other cultures). What happen if a cell was climatically suitable for different human groups? The model can exclude the Neanderthal from accessibility equation?

For all these reasons, I cannot recommend the manuscript for publishing and I’m forced to reject this paper again.

Reviewer #3:

Remarks to the Author:

The authors have addressed most of the concerns raised in my original review. The expanded sections on model structure and validation offer greater clarity, Improvements in the quality and readability of key figures are evident throughout, and the authors thoughtfully acknowledge major data limitations while in their approach to model calibration and validation. The authors are also to be commended for making their code and data publicly available to enhance future reuse and reproducibility of this analysis.

The inclusion of additional baseline simulations in particular has strengthened the manuscript, making the impacts of various modeling decisions more clear and accessible to the reader. If space allows, consider defining each simulation experiment in the caption of Figure 9 directly, in addition to the existing descriptions in Table 4 and the text, to make this analysis even more clear and impactful.

My earlier reservations about the absence of spatial cross-validation remain, however. The downsampling of nearby sites for some parts of the analysis is a good first step, but the the irregular spatial sampling of sites (cited here as a reason not to use spatial cross validation) is the exact reason why a such validation is needed — this heterogeneity potentially impact all analyses based on these data and it is risky to ignore it. However, this is a relatively minor point in light of the other efforts made to refine the validation of the model, and the limitations of the dataset are made clear throughout; therefore, I don't consider it a sticking point for publication.

Reply to Review Reports (Author's reply is marked purple).

Reviewer #1 (Remarks to the Author):

We thank Reviewer #1 for the constructive comments, which obviously involved very deep thinking. We are also very interested in the questions Reviewer #1 has raised and will devote future research efforts to answer some of them. We realize that we are working in field of considerable uncertainties. The core issues being addressed in our paper on the dispersal of modern humans in Europe have been under heavy debate over decades and colleagues from different backgrounds have very different and often strong views. As Reviewer #1 has pointed out, just as our paper is being reviewed, several new studies have been published, plus one “that will cause discomfort to some pre-historians”. So, “discomfort” seems to be a norm.

As we have replied in our 1st revision, the IUP/NEA and AUR human interactions are particularly tangled and hugely uncertain. From the modeling perspective, we are seriously limited by the lack of IUP data for the derivation of the Human Existence Potential (HEP). Unless we have a sufficient understanding how humans of NEA/IUP and AUR interacted, it will be difficult to parametrize and quantify the interactive processes and our model results will end up in controversies. Given the state of the art, we believe an answer with confidence to the IUP influences on AUR dispersal in Europe is beyond the scope of this paper. Nevertheless, our IUP experiments, stimulated by the first review report of Reviewer #1, are still useful as a first numerical/model try to address the concerns raised by Reviewer #1. These results should be interpreted in conjunction with the assumptions made.

The IUP experiment added to the resubmission is apparently to partially address concerns about the omission of this important phase of modern human dispersal. I'm not sure that it strengthens the manuscript because it raises additional concerns and questions. First, I wonder how the results would look if Uluzzian sites from Italy and Neronian sites from southern France were included as modern human associated IUP.

Following the suggestions of the Reviewer #1, we added two new experiments in the revised version. In Experiment BS10, we examine what could happen if modern humans were originated from the Uluzzian sites in Italy and CHÂT sites in France. The outcomes of these experiments are discussed.

Second, why should the birthrate for these modern humans be 1/10th that of Aurignacian modern humans? If there really has emerged a consensus that demographic factors explain the demise of Neanderthals and success of modern humans (Vaesen et al. 2021), why did these modern humans not have the same reproductive capacity as Aurignacian modern humans? Is it due to hybridization and possibly lowered reproductive success? The limited skeletal evidence dated to the IUP-Aur-P1 timeframe shows that these earlier modern humans had recent Neanderthal ancestry suggesting that the demographic assimilation of Neanderthal populations and their widespread replacement was taking place during this period. Neanderthals were almost entirely gone by the Aur-P2 so what we may be looking at is a replacement and assimilation of the earlier modern human lineage by a subsequent one, as explicitly implied by the authors' climate model.

We agree with the views of Reviewer #1 in that “limited skeletal evidence dated to the IUP-Aur-P1 time frame shows that these earlier modern humans had recent Neanderthal ancestry suggesting that the demographic assimilation of Neanderthal populations and their widespread replacement was taking place during this period” (Hajdinjak et al. 2021, Initial Upper Palaeolithic humans in Europe had recent Neanderthal ancestry. *Nature* 592, 253–257, <https://doi.org/10.1038/s41586-021-03335-3>). Based on the review by Bergström et al. (2021, Origins of modern human ancestry. *Nature* 590, 229–237. <https://doi.org/10.1038/s41586-021-03244-5>), NEAs contribute to 2% of the genetic signals of the modern humans, suggesting that interactions between NEAs and AMHs did take place,

but the scale of the interactions was limited. In addition, we have to bear in mind that the timing and intensity of this process have regional differences. At the time, as humans of the Aurignacian arrived in Europe, the NEA population was under decline, if not already extinct. The Iberian Peninsula has been long considered to be the last refuge of the NEAs. However, the timing of NEA extinction in Iberia is also highly debatable. The NEA fossils from El Sidron are dated directly to 48.4 pm 3.2 ka cal BP (Highham et al. 2014). The fragmentary NEA remains from Sima de las Palomas de Cabezo Gordo were found together with burnt fauna bones radiocarbon dated to 42.01 and 38.4 ka cal BP. The radiocarbon dating of bones is less reliable and it cannot be excluded that their accumulation occurred after the NEAs disappeared. More recent dating suggests that most of the sites dated to the late MP in Iberia were abandoned by the NEAs probably before ca. 45 ka. This indicates a decline of the NEA population already in Heinrich Event 5 and the eventual disappearance at the latest before HE4. Therefore, it is very uncertain, whether there was a significant population overlap between NEA and AMH of AUR in Europe.

It is difficult to estimate what the IUP population density in Europe might be at the time as humans of the AUR arrived in Europe around 43 ka cal BP. The climate from about 47 ka cal BP to 43.5 ka cal BP (Greenland Interstadial 12 to Stadial 12) was cooling, reaching the coldest phase as humans of the AUR appeared in east Europe. It is thus plausible to argue that the IUP population probably had a density too low to significantly alter the courses of the AUR expansion, as our Experiment BS8 suggests.

Our model has an internal logic for the population growth and dispersal, if the population growth is too low, then rapid human dispersal is not possible. The population growth rate depends on the cultural carrying capacity, an external model parameter, which represents the capacity of humans to harness resources in a given environment. While IUP and AUR were all AMHs, the different cultures had different carrying capacities which in turn lead to different population growth rates. If we make the scenario that the IUP cultures had the same HEP and the same cultural carrying capacity as the AUR, then the model would predict stronger expansion of the IUP cultures before the arrival of humans of the AUR. This is however not we know from archeology because the IUP cultures did not seem to have resulted in large scale firm settlement.

The suggestion that "so what we may be looking at is a replacement and assimilation of the earlier modern human lineage by a subsequent one" is an interesting proposition. Results of BS8 and the low palaeogenetic input of IUP humans to the later AMHs in Europe on the other side allow to assume instead a failed colonization attempt of the IUP humans. A failure that could be related to climatic changes or a low population density or a combination of the two. It is interesting to note here that the comparison of AUR-P1 and AUR-P2 shows that humans of AUR-P1 had a lower adaptability than of AUR-P2. If we look at the situation during the LGM in Central Europe, which was largely abandoned by humans, it is clear that the adaptive capacity of AHMs still had limits even at this late stage.

As Reviewer #1 seems to be suggesting, it is highly interesting to test whether the IUP cultures evolved into a culture similar to the Aurignacian, which led to large scale settlement in Europe. This is possible and this hypothesis, if confirmed, would turn upside down our view on AMH expansion in Europe (may be elsewhere). But to thoroughly test this hypothesis, we need to do a lot more work.

Following the recommendation of Reviewer #1, we conducted Experiment BS9, which is a repeat of Experiment BS8, with the model parameters (e.g., birth rate, drift and diffusion parameters) as set for the humans of the AUR. We discussed the results of BS9.

Having said that I think the reorganization of section 3.5 was helpful to clarify aspects of the model.

Line 061: should be Mandrin, not Mandarin
Thanks. Corrected.

Line 980: “Little Asia?” Is that Asia Minor? Maybe just continue to use Anatolia.
Thanks. Corrected.

I still have issues with the authors’ assumptions regarding the spread of modern humans.

1) Aurignacian represents dispersal/migration/colonization of empty landscapes by modern humans. Equally plausible explanation for such a unified package is the adoption of it through existing exchange networks.

Again, this is a very interesting proposition. It is certainly a plausible explanation of the Aurignacian as an adoption through existing exchange networks. We believe AUR-P1 is better explained as a dispersal/migration/colonization process, based on the chronology of the first Aurignacian sites in east Europe, west Europe and Iberia Peninsula. This chronology shows the different stages of the dispersal, as discussed in our paper. Indeed, the observed chronology and the propagation routes of the culture are important measures for the model performance. However, AUR-P2 is indeed better explained as an adoption/expansion process. As shown in Figure 1 of the paper (see also Shao et al. 2020, Human-existence probability of the Aurignacian techno-complex under extreme climate conditions. *Quat Sci Rev* 263, <https://doi.org/10.1016/j.quascirev.2021.106995>), the GS10-GS9/HE4 cold period forced the Aurignacian settlement areas to contract in the period around 40 - 38.5 ka cal BP. The resettlement is better explained as an adoption/expansion of the population networks, mostly clearly seen at the northern boundary of the techno-complex and in Iberia.

We believe the views of the Reviewer #1 are worth of pursuing and will conduct a dedicated new study in the near future to test the propositions.

2) IUP at Bacho Kiro known largely from old excavations but recent excavations demonstrated the age and integrity of lithic assemblages.

Recent work at Bacho Kiro has substantially expanded our knowledge. But as it is so often the case with detailed modern excavations in sites with a long research history and a lot of ancient recovered material, the available modern database is small in frequency and the connection to the ancient samples is not always straightforward.

3) Grotte Mandrin represents the only place where a hominin is associated with the Neronian. At Abri du Maras, the Neanderthal association is assumed despite a lack of hominin remains, including bones, sedaDNA, and proteomes. The Mandrin finds cannot be dismissed through contrast with Maras.

Concerning Grotte Mandrin we agree with Ruebens et al. (2022) that we are still awaiting in depth studies of the whole ensemble and stratigraphy of the site to fully understand the Neronian and its cultural and anthropological attribution.

4) The Oase 1 specimen, dated 42-37 ka calBP, once served as the best evidence to associate the Aurignacian with AMH. Fu et al. (2015) showed that it had recent Neanderthal ancestry and wasn’t closely related to later Europeans. Hublin et al. (2020) suggested that it might be older because the date was produced using antiquated pretreatment protocols. Posth et al. (2023) place the oldest modern human closely related to subsequent Europeans at Kostenki ~37 ka calBP and Goyet ~35 ka calP. Thus, it is plausible that the Aurignacian itself represents multiple migrations of people, some of which were unsuccessful in the sense that they did not contribute genetic heritage to existing

European lineages. The authors' Aur-P1 may be no more representative than any dismissed IUP migrations. Aur-P2 would be the first truly successful (at least in terms of impact on subsequent Europeans) modern human dispersal into Europe. This scenario is explained by the authors' demographic estimates for a major population decline between Aur-P1 and Aur-P2.

Given the uncertain radiocarbon dating of the Oase specimen that might be too young and found without any archaeological context, it might belong to these early unsuccessful attempts of colonizing central Europe from the Southeast. Given the AUR-P1 we find another situation concerning site frequencies, dispersal and technology. This makes it very different from the IUP whose cultural signal is currently so weak that it allows a wide range of interpretations.

Some developments since submission that might influence the authors' thinking about the Chatelperronian:

Gicqueau, A., Schuh, A., Henrion, J., Viola, B., Partiot, C., Guillon, M., Golovanova, L., Doronichev, V., Gunz, P., Hublin, J.-J., Maureille, B., 2023. Anatomically modern human in the Châtelperronian hominin collection from the Grotte du Renne (Arcy-sur-Cure, Northeast France). *Scientific Reports* 13, 12682.

Henrion, J., Hublin, J.-J., Maureille, B., 2023. New Neanderthal remains from the Châtelperronian-attributed layer X of the Grotte du Renne (Arcy-sur-Cure, France). *Journal of Human Evolution* 181, 103402.

We did not cite these papers in the revised versions, as we need to have more time to work through their studies to place them in the context of our work.

Plus, a third that will cause discomfort to some prehistorians:

Slimak, L., 2023. The three waves: Rethinking the structure of the first Upper Paleolithic in Western Eurasia. *PLOS ONE* 18, e0277444.

The latest paper by Slimak (2023) is a large scale study covering a space of nearly 4.000 km from east to west using stone tool typology. It is a challenging study that attempts to integrate a very small number of spatially isolated sites into three successive dispersal waves. We find it inspirational but believe the details of this study need to be scrutinized. The greatest gain of this study is to raise the awareness that we should expect several colonization attempts of AMH in Europe before the Aurignacian, which apparently failed. The essential approach of Slimak (2023) coincides with our macroscopic considerations, but our approach is dynamic-model and data combined, which gives the model backing to the data analysis of the transition processes from the Middle Palaeolithic to the Upper Palaeolithic. We are in the process of carrying out a new detailed study in this direction to investigate the advances and retreats of populations.

In Section 3.5.2 IUP Experiments, we have added a substantial section on this topic to present our views. The work of Slimak (2023) is referenced.

Reviewer #2 (Remarks to the Author):

Dear editor,

thank you for giving me the possibility to review this manuscript.

In this R2 of the manuscript “Reconstruction of Human Dispersal during Aurignacian on Pan-European Scale”, the authors increased the discussions about the most critical points in the main text. However, I was not convinced that they gave significant improvements to the manuscript. The authors did not solve my skepticism about the articles.

First, I asked more details about the HEP models published elsewhere. The authors have done a simple copy and paste of several paragraphs. Consequently, it is very unclear what the authors have really done in their manuscript and/or what they have retrieved from the previous model calibration setup.

We have been working as a group on the HEP model for almost 10 years. As stated in Lobo et al. (2010), there are different techniques to estimate the niche for human existence. Here we opted to use multivariate second order logistic regression, although we have used similar method as Lobo et al. (2010) in one of our earlier studies (Maier et al. 2016, Demographic estimates of hunter–gatherers during the Last Glacial Maximum in Europe against the background of palaeoenvironmental data. *Quat Int* 425, 49-61, <https://doi.org/10.1016/j.quaint.2016.04.009>). The methodology we used here has been carefully tested, and based on our validation and sensitivity analysis, we are satisfied that the HEP results are robust. The methodology, validation and HEP results have been published in three papers:

1. Klein et al. 2021: Human existence potential in Europe during the Last Glacial Maximum, *Quat Int* 581–582, 2021, 7-27, <https://doi.org/10.1016/j.quaint.2020.07.046>.
2. Shao et al. 2021: Human-existence probability of the Aurignacian techno-complex under extreme climate conditions, *Quat Sci Rev* 263, 106995, <https://doi.org/10.1016/j.quascirev.2021.106995>.
3. Klein et al. 2023: Assessing climatic impact on transition from Neanderthal to anatomically modern human population on Iberian Peninsula: a macroscopic perspective. *Sci Bull* 68, 1176-1186, ISSN 2095-9273, <https://doi.org/10.1016/j.scib.2023.04.025>.

In the peer review processes of these paper, various questions (drop sites, add new sites, empty areas, data uncertainties, etc.) have been considered and tested. We have intensively discussed these issues with the referees. Therefore, we felt we have said enough about the HEP issues and are somewhat surprised by the comments of Reviewer # 2. As much of the work has been published elsewhere, we are not sure how much details we must repeat in this Nature/Comm paper which is supposedly concise. However, to meet the demands of Reviewer # 2, we have now expanded the section, with more details, regarding the HEP calculations.

For instance, Shao et al. 2021 selected 4 uncorrelated predictors: BIO1, BIO4, BIO13, and BIO15 and in the current paper, I read: “The selection the predictors requires they are not collinear and allow the logistic regression to distinguish the presence/absence records. We use the hierarchical correlation cluster technique to examine the collinearity between the candidate predictors. Shao et al(2021b) showed that Tm (annual mean temperature), Tδ (temperature seasonality), Rm (annual rainfall) and Rδ (rainfall seasonality) are the most suitable predictors for the Palaeolithic humans.” Although, I assume that the HEP was already calibrated and validated in Shao et al. 2021, the authors listed a different list of predictors since Rm /BIO 12 (“annual rainfall”) was never selected by Shao and colleagues.

We thank Review # 2 for pointing this inconsistency in the name list of the predictors. There is a typo in the manuscript, instead of BIO12, BIO13 is the correct BioClimate variable as used in Shao et al. (2021). We have now corrected the typo. But please note that, as BIO12 and BIO13 belong to the same group of variables of strong co-linearity (with correlation separation less than 0.1, as the figure below shows), the use of BIO12 and BIO13 makes no difference in the HEP results.

Figure 1: Hierarchical correlation cluster of the 17 bioclimatic variables. The threshold for correlation separation is set to 0.4 (from Shao et al. 2021).

As further clarification, I would have liked to know what it means a “subset of the grid cells where no archaeological site exists as “absence cells”. There is a lot of literature about the number and the appropriate locations of absence points used to calibrate a model (Barbet-Massin et al. 2012) but the authors did not give any details about it. Furthermore, I would point out to the authors that they used the human fossil records to define the empty cells. Consequently, they cannot rely on the “true” absence points but “pseudo-absence” datapoints instead (Lobo et al. 2010) because of the dataset limitations.

- Barbet-Massin, M., Jiguet, F., Albert, C. H., & Thuiller, W. (2012). Selecting pseudo-absences for species distribution models: How, where and how many?. *Methods in ecology and evolution*, 3(2), 327-338.
- Lobo, J. M., Jiménez-Valverde, A., & Hortal, J. (2010). The uncertain nature of absences and their importance in species distribution modelling. *Ecography*, 33(1), 103-114.

As stated above, we are aware of the different ways for computing the HEP. We could have followed the methods suggested by Lobo et al. (2010), and indeed, a similar work has been done by the senior author, as reported in Maier et al. (2016). We have now cited the paper of Lobo et al. (2010), but not the paper by Barbet-Massin (2012) which we have difficulty to understand. In our study, we favored using multivariate second order logistic regression which, in our view, is a more efficient and compact method. It is particularly advantageous for calculating the HEP temporal variations over a time period of several thousand years. The HEP model trained with archeological data interpolates the probability of human existence using climate predictors.

For HEP calculation, the handling of the archeological site data is as follows:

1. The study domain is divided into $I \times J$ cells, with a horizontal resolution of ~ 50 km;
2. Cells over large water bodies (e.g. ocean) are excluded from consideration

3. Cells in which archeological sites are found are defined as presence cells. If a cell contains more than one site, only one site is counted, to reduce data redundancy
4. The rest cells are “absence cells” which consist of “a-priori absence cells” and “pseudo-absence cells”. A-priori absence cells are cells where human presence was impossible according to our pre-knowledge. A set of criteria is used to define the a-priori absence cells, including:
 - a) BIO1 (Annual mean temperature) below -2°C or above 16°C ;
 - b) BIO13 (Rainfall in wettest month) below 30 mm or above 250 mmThese values correspond to a 95% confidence level for human existence, estimated based on the PDFs of the relevant variables from the archeological sites.

The rest of the absence cells are “pseudo-absence cells”. They have three different interpretations: (1) regions humans existed, but no human records were not found (site not preserved, region not studied, region studied but no records were found); (2) regions humans could exist, but unexplored; and (3) regions where climate/environmental conditions were unsuitable for human existence. This classification of absence-cells is fully consistent with the “methodological”, “contingent” and “environmental” absence listed by Lobo et al. (2010).

No general statement can be made about the nature of the pseudo-absence cells, but three options can be considered:

- (1) Pseudo-absence cells do not enter into the HEP model training. The HEP values for these cells are calculated by using the HEP model trained with the presence and the a-priori absence records;
- (2) Methodological, contingent and environmental absence are equally possible and one third of the pseudo-absence cells can be treated as absence cells;
- (3) The probability of finding archeological record among pseudo-absence cells is identical to that of finding an archeological record reflected in the existing data set, namely, $N_p/(N_p + N_{ps})$, where N_p is the total number of presence records and the N_{ps} is the total number of pseudo-absence records.

In this study, we opted to take option 3 which is the most stringent option (making HEP smaller than the other two options). We added above information to the revised paper.

Second, I asked to verify for spatial correlation in native data by using another cross-validation scheme. The authors replied to me: “We have also thought about the “spatial block validation method” (e.g., Roberts et al. 2017: Cross validation strategies for data with temporal, spatial, hierarchical, or phylogenetic structure. *Ecography* 40, 913–929), but this method may not be suitable for our study, as it leads to large variations in the HEP patterns due to the fact that the neither the “presence” records (the number of which is small) no the climate data are spatially homogeneously distributed.” The authors did not follow my suggestion and with the last sentence, they practically admitted their dataset is spatially biased.

We are seriously puzzled by this comment of the referee and how the conclusion is reached that “they practically admitted their data set is spatially biased”. In particular, we do not understand what “spatially biased” means.

We go back to the very question what the HEP model is about. HEP is introduced to define the climate conditions under which humans lived. Ideally, we need an archeological site data set which covers all climate conditions. The archeological site data set we used is the only full and quality-checked data set our community has. Thus, it is not meaningful to ask whether the data set is sampled with spatial bias. Rather, we should ask whether the data set represents the totality of the climate conditions.

However, the climate conditions under which humans of the AUR lived varied greatly in space. If we chose a subset of the data blocked to a region, we can derive a HEP model, but we cannot expect this model describes also the climate conditions in a different region. In Shao et al. (2021), we have shown that AUR-P1 and AUR-P2 have different HEPs, because AUR-P1 sites were located in a somewhat different climate regime as AUR-P2, by comparing the differences, we can estimate the human adaptation to climate (Figure 2)

Figure 2: HEP and HEP difference (Aur-II – Aur-I). For interstadial times (a) HEP for Aur-I; (b) HEP for Aur-II; (c) HEP difference, and for stadial times (d) HEP for Aur-I; (e) HEP for Aur-II; and (f) HEP difference.

In Maier et al. (2016), we separated east Europe from west Europe and found different climate adaptations (Figure 3), but clearly we cannot use the east Europe data to validate the west Europe model, due to the fact that climate is spatially heterogeneous, not because the archeological site data is “spatially biased”.

Figure 3: A data divided into an east Europe data set and a west Europe data set. The resulting HEP derived from the west Europe model and the east Europe model.

Because our aim is to estimate the climate conditions for the whole culture, it is not advantageous to re-sample the archeological site data using spatial blocks. By the way, how large should spatial blocks be? The choice of different spatial block sizes would lead to very different results. Therefore, the random reduction of the archeological sites for HEP model training is a best way we can think of.

If we select a spatial block which is much smaller than the domain (indeed this is what random reduction implies, a spatial block is namely a cell), then the HEP model results would not be influenced very much.

Eventually, the OWM has two components, the HEP model (a machine-learning procedure for determining the spatial-temporal variable of HEP) and the dynamic human dispersion model (HDM) whereby the first has already calibrated in Shao et al. 2021. Hence, the “new” contribution of the authors for OWM is limited to the HDM and accessibility parameter. Anyway, I strongly suspect they created an overestimation of human density (restricted to Aurignacian only) because their OWN does not take into account the interaction with other human groups (Neanderthals and human with other cultures). What happen if a cell was climatically suitable for different human groups? The model can exclude the Neanderthal from accessibility equation?

It is correct to say that the main contribution of this paper is the HDM, which is not trivial. To the best of our knowledge, this is the first numerical reconstruction of the Aurignacian dispersal on the pan-European scale, and it is the first example of model-data (in terms of climate HEP, accessible HEP and available HEP) coupled approach to human dispersal. These achievements have far-reaching influences on the development of the research field.

Our research community does not know the size of the Aurignacian population. Studies on the Aurignacian population size are very few. Reviewer # 2 suspects that we overestimated human density, we do not know on what basis this assessment is made. We have already stated in the text that “for Europe, Bocquet-Appel et al. (2005) reported a population size of about 4000 with a 95% confidence interval of [1700, 37000]. Our model-estimated Lambda maximum is about two times the upper limit reported in Bocquet-Appel et al. (2005). The main reason for the discrepancies lies in the high-hep-no-site areas (i.e., areas of high human existence potential but with no archaeological sites found), for which our model may predict human presence while demographic estimates based on archaeological data only assume zero or very low human density”. Therefore, we do not agree with the statement that our estimates of population density are overestimates, instead they are the most rational estimates we can make based on all of the means at disposal and they need to be further scrutinized by future independent studies.

It is very nicely stated in Lobo et al. (2010), “A great part of the modeling exercises overlook the conceptual and methodological implications of discerning between potential and realized distribution.” While the HEP model gives the potential, the HDM gives the realized distribution, and this combination is the essence of our contribution which remedies precisely the “overlook” Lobo et al. (2010) pointed out..

With regard to the NEAs and IUP humans, we have presented our views in our replies to Reviewer # 1 and in the revised paper. There are two problems here:

- (1) Because of the very limited data we have for the NEA/IUP populations, we are for now not in the position to produce reasonable HEP estimates for these human populations;
- (2) Because we do not have sufficient knowledge about how NEA/IUP interacted with AMHs, to parameterize such interactions would be highly speculative and would lead to controversies.

For these reasons, as already repeatedly stated in the revised paper, we believe it is beyond the scope of this study to satisfactorily answer the questions surrounding NEA/IUP.

As we replied to Reviewer #1: “Based on the review of Bergström et al. (2021, Origins of modern human ancestry. *Nature* **590**, 229–237. <https://doi.org/10.1038/s41586-021-03244-5>), NEAs contributes to about 2% of the genetic signals of the modern humans, suggesting that interactions between NEAs and AMHs took place, but the scale of the interactions was limited.

It is difficult to estimate what the IUP population density in Europe might be at the time as humans of the AUR arrived in Europe around 43 ka cal BP. The climate from about 47 ka cal BP to 43.5 ka cal BP (Greenland Interstadial 12 to Stadial 12) was cooling, reaching the coldest phase as humans of the AUR arrived in east Europe. It is thus plausible to argue that the IUP populations probably have such a density which is too low to significantly alter the courses of AUR expansion.

Reviewer #2 asks specifically “What happen if a cell was climatically suitable for different human groups?” Please note that the model has an internal logic for human dispersal and population growth. If the population growth is too low, then human dispersal is unlikely. We emphasize that the population growth rate depends not only on environmental and climate conditions, but also on the cultural carrying capacity, a culture-dependent external parameter representing the human capacity to harness resources. If a cell was climatically suitable for different human groups, then growth rate of the human groups will be dependent on the relevant cultural carrying capacities. The total population will create a population pressure (Figure 6 in our study) which will impact differently on the different population groups. We cannot speculate on other possible processes, such as conflict between the groups. At the time humans of the Aurignacian emerged in Europe, the other humans groups come into consideration are humans of the IUP techno-complexes. The role of NEAs is uncertain, but as we have argued, and indeed show in Klein et al. (2023), their extensive coexistence with the AMHs of the Aurignacian is

unlikely. Therefore, to the specific question “The model can exclude the Neanderthal from accessibility equation?”, our view is “yes” under plausible assumptions.

We would like to draw Reviewer #2’s attention to the experiments we have done in relation to the IUP (BS8, BS9 and BS10). The model results suggest that, by the time humans of the Aurignacian arrived in Europe, the populations of the IUP cultures were so low to produce a substantial alteration to the expansion of the Aurignacian, because on the macroscopic scale, there was no serious competition for resources between the different species and between the humans of the IUP and AUR. Of course, we cannot rule out the conflicts and interactions which might cause local fluctuations.

For all these reasons, I cannot recommend the manuscript for publishing and I’m forced to reject this paper again.

We have tried our best to address the concerns raised by Reviewer #2 and very much hope the Reviewer recognizes the value of this study to the progress of the entire research field. The model-machine-learning coupled approach is at early stage of development, we therefore do not claim to have solved all problems in a single paper. Rather, the merit of the paper should be judged whether it represents a significant progress of the research field. We are certainly convinced that we have.

Reviewer #3 (Remarks to the Author):

The authors have addressed most of the concerns raised in my original review. The expanded sections on model structure and validation offer greater clarity, Improvements in the quality and readability of key figures are evident throughout, and the authors thoughtfully acknowledge major data limitations while in their approach to model calibration and validation. The authors are also to be commended for making their code and data publicly available to enhance future reuse and reproducibility of this analysis.

We are grateful to Referee # 3 for the encouragement.

The inclusion of additional baseline simulations in particular has strengthened the manuscript, making the impacts of various modeling decisions more clear and accessible to the reader. If space allows, consider defining each simulation experiment in the caption of Figure 9 directly, in addition to the existing descriptions in Table 4 and the text, to make this analysis even more clear and impactful.

We followed the suggestion of the Referee # 3.

My earlier reservations about the absence of spatial cross-validation remain, however. The downsampling of nearby sites for some parts of the analysis is a good first step, but the the irregular spatial sampling of sites (cited here as a reason not to use spatial cross validation) is the exact reason why a such validation is needed — this heterogeneity potentially impact all analyses based on these data and it is risky to ignore it. However, this is a relatively minor point in light of the other efforts made to refine the validation of the model, and the limitations of the dataset are made clear throughout; therefore, I don't consider it a sticking point for publication.

We have presented our view on this issue in our lengthy reply to Review # 2.

Reviewers' Comments:

Reviewer #1:

None

Reviewer #2:

None

Reviewer #3:

Remarks to the Author:

The latest revision of this manuscript continues to exhibit issues in its approach to model validation, a concern that has been previously highlighted by myself and (more emphatically so) by Reviewer Two. The authors' response and adjustments in this revision, arguing against the use of spatial cross validation, unfortunately deepen these concerns rather than addressing them.

The fundamental role of spatial cross-validation in predictive spatial modeling is to assess the robustness and generalizability of the model across different spatial domains. The purpose is not, as the authors appear to misconstrue, to use one or more of the models trained during cross-validation for prediction. Rather, it is to evaluate the *entire modeling procedure* to ensure reliable performance when applied to new data. That the model yields different predictions when trained on different spatial subsets of the data (as was the case for the east/west HEP models the authors cite) is exactly why this is necessary, as it helps us quantify how sensitive the predictions are to this spatial heterogeneity. Similarly, the uncertainty they note in how large to make the spatial blocks further supports this point — sensitivity to block sizes should be assessed empirically (see references to “h-block” cross validation in palynology) or rooted in a priori knowledge of the scale of spatial autocorrelation in the data. The author's emphasis on adequately sampling climate space, not just geographical space, is reasonable, but these same principles still apply — any source of structure in the dataset not accounted for during cross validation will lead to an overestimate of the model's performance on unseen data.

Part of this issue seems to arise from a misunderstanding of what one does with the results of cross validation. The authors seem to suggest that one of the models trained during the spatial cross validation would need to be chosen as the “final model,” or perhaps that the predictions from the models trained on each fold would be averaged together. This is not the case. Cross validation is used to assess the modeling procedure (preprocessing, variable and hyper parameter selection, etc), not a particular fitted model. That is, after a model pipeline is cross validated, the individual model fits to each (spatial) fold can be discarded and the final modeling pipeline applied to the full training dataset.

This misunderstanding is further compounded in their description of what they refer to as “bootstrap cross-validation”, which, as described here, sounds like cross validation through repeated train-test splits (bootstrapping, on the other hand, means to sample from the original data with replacement to generate new datasets, but it is not clear if that was done here and if so would have further consequences for how we assess model performance). In earlier work on HEP by the authors, they describe a similar approach where they average the predictions from each of the 1,000 models fit on bootstrapped data — this is a valid but distinct approach known as bootstrap aggregation or “bagging,” but it is not clear if the same method was employed here. The lack of clarity on what exactly was done makes it difficult to assess the validity of the HEP predictions on which the rest of the analysis relies.

Successful (cross) validation is imperative for ensuring that the confidence in the model's predictions is well-founded. This involves testing the entire modeling process, including data preprocessing and variable selection, within each cross-validation fold. This step is crucial to prevent data leakage and overfitting. In the authors' current methodology, these processes appear to be conducted prior to the division of data, which can lead to misleading results. A clear case of this that has emerged in this revision is the generation of “a priori absences” to train the HEP model. If I understand correctly, the authors used the empirical distribution of sites to establish maximum and minimum temperature and

precipitation thresholds, generated “true absence” points in locations that exceeded those thresholds, then trained the HEP model to distinguish the two (as well as additional pseudo-absences). If so, this approach is inherently problematic as it biases the model training process by incorporating knowledge about the data distribution beforehand — it essentially 'peeks' at the data before model training and leads the resulting model to detect the patterns the authors have implicitly imposed on it. To maintain the integrity of the model, such steps need to be embedded within each fold of the cross-validation to avoid data leakage and ensure unbiased evaluation.

Despite the efforts in revising the manuscript, the authors' approach to (spatial) cross-validation remains inadequately addressed. The current methodology undermines the reliability of their model and its applicability to new datasets, and recent revisions have only made these methods less clear. As mentioned in my previous review, I am sympathetic to the to the argument that the current manuscript is focused on the stochastic simulation model, not the HEP data used as inputs, but there are still too many uncertainties in the latter to adequately assess the performance in the former.

I suggest the authors pursue one of three options:

- Replace the “bootstrapping” approach with a spatial cross-validation routine. “Spatial-block” CV is only one option, and there are others that may be more appropriate here (see <https://spatialsample.tidymodels.org/> for examples in R, and particularly the implementation in tidySDM package <https://doi.org/10.1101/2023.07.24.550358>).
- Assess the degree to which the HEP model is extrapolating into entirely new socio/environmental contexts or just interpolating the observations in the training data. The concerns with cross validation are primarily focused on assessing model failures during extrapolation, and an assessment of where/when extrapolation is actually happening might be useful (see <https://applicable.tidymodels.org/> for examples in R).
- Use a minimalist climate-potential model in place of the full HEP. For example, if you do have good a priori knowledge of upper and lower temperature thresholds for human habitation, why not just use that as a simple “envelope” model for the climatic constraint on potential human activity and let the OWM and other terrain/environmental submodels do the rest?

Reply to Review Reports (Author's reply is marked purple).

Reviewer #3 (Remarks to the Author):

The latest revision of this manuscript continues to exhibit issues in its approach to model validation, a concern that has been previously highlighted by myself and (more emphatically so) by Reviewer Two. The authors' response and adjustments in this revision, arguing against the use of spatial cross validation, unfortunately deepen these concerns rather than addressing them.

The fundamental role of spatial cross-validation in predictive spatial modeling is to assess the robustness and generalizability of the model across different spatial domains. The purpose is not, as the authors appear to misconstrue, to use one or more of the models trained during cross-validation for prediction. Rather, it is to evaluate the entire modeling procedure to ensure reliable performance when applied to new data. That the model yields different predictions when trained on different spatial subsets of the data (as was the case for the east/west HEP models the authors cite) is exactly why this is necessary, as it helps us quantify how sensitive the predictions are to this spatial heterogeneity. Similarly, the uncertainty they note in how large to make the spatial blocks further supports this point — sensitivity to block sizes should be assessed empirically (see references to “h-block” cross validation in palynology) or rooted in a priori knowledge of the scale of spatial autocorrelation in the data. The author's emphasis on adequately sampling climate space, not just geographical space, is reasonable, but these same principles still apply — any source of structure in the dataset not accounted for during cross validation will lead to an overestimate of the model's performance on unseen data.

Part of this issue seems to arise from a misunderstanding of what one does with the results of cross validation. The authors seem to suggest that one of the models trained during the spatial cross validation would need to be chosen as the “final model,” or perhaps that the predictions from the models trained on each fold would be averaged together. This is not the case. Cross validation is used to assess the modeling procedure (preprocessing, variable and hyper parameter selection, etc), not a particular fitted model. That is, after a model pipeline is cross validated, the individual model fits to each (spatial) fold can be discarded and the final modeling pipeline applied to the full training dataset.

This misunderstanding is further compounded in their description of what they refer to as “bootstrap cross-validation”, which, as described here, sounds like cross validation through repeated train-test splits (bootstrapping, on the other hand, means to sample from the original data with replacement to generate new datasets, but it is not clear if that was done here and if so would have further consequences for how we assess model performance). In earlier work on HEP by the authors, they describe a similar approach where they average the predictions from each of the 1,000 models fit on bootstrapped data — this is a valid but distinct approach known as bootstrap aggregation or “bagging,” but it is not clear if the same method was employed here. The lack of clarity on what exactly was done makes it difficult to assess the validity of the HEP predictions on which the rest of the analysis relies.

Successful (cross) validation is imperative for ensuring that the confidence in the model's predictions is well-founded. This involves testing the entire modeling process, including data preprocessing and variable selection, within each cross-validation fold. This step is crucial to prevent data leakage and overfitting. In the authors' current methodology, these processes appear to be conducted prior to the division of data, which can lead to misleading results. A clear case of this that has emerged in this revision is the generation of “a priori absences” to train the HEP model. If I understand correctly, the authors used the empirical distribution of sites to establish maximum and minimum temperature and precipitation thresholds, generated “true absence” points in locations that exceeded those thresholds, then trained the HEP model to distinguish the two (as well as additional pseudo-absences). If so, this approach is inherently problematic as it biases the model training process by incorporating knowledge about the data distribution beforehand — it essentially ‘peeks’ at the data before model training and leads the resulting model to detect the patterns the authors have implicitly imposed on it. To maintain the integrity of the model, such steps need to be embedded within each fold of the cross-validation to avoid data leakage and ensure unbiased evaluation.

Despite the efforts in revising the manuscript, the authors' approach to (spatial) cross-validation remains inadequately addressed. The current methodology undermines the reliability of their model and its applicability to new datasets, and recent revisions have only made these methods less clear. As mentioned in my previous review, I am sympathetic to the to the argument that the current manuscript is focused on the stochastic simulation model, not the HEP data used as inputs, but there are still too many uncertainties in the latter to adequately assess the performance in the former.

I suggest the authors pursue one of three options:

- Replace the “bootstrapping” approach with a spatial cross-validation routine. “Spatial-block” CV is only one option, and there are others that may be more appropriate here (see <https://spatialsample.tidymodels.org/> for examples in R, and particularly the implementation in tidy SDM package <https://doi.org/10.1101/2023.07.24.550358>).
- Assess the degree to which the HEP model is extrapolating into entirely new socio/environmental contexts or just interpolating the observations in the training data. The concerns with cross validation are primarily focused on assessing model failures during extrapolation, and an assessment of where/when extrapolation is actually happening might be useful (see <https://applicable.tidymodels.org/> for examples in R).
- Use a minimalist climate-potential model in place of the full HEP. For example, if you do have good a priori knowledge of upper and lower temperature thresholds for human habitation, why not just use that as a simple “envelope” model for the climatic constraint on potential human activity and let the OWM and other terrain/environmental submodels do the rest?

We thank Reviewer #3 for his/her effort for providing us with the constructive comments, which motivated us to think more deeply into our work. The key issue raised by the referee is the validation of the human existence potential (HEP). While we thought we have addressed this issue in our previous revisions and lengthy replies, there does seem to be points which need clarification. In this revision, we have followed the recommendations of the referee and provided additional assessment on the performance of the HEP model, and we have carried out additional numerical experiments. We also feel that there are several points which seem to be misunderstood. To these points, we present our views.

Much of the confusion and the concerns raised by the referee(s) seem to be related to the technique how archaeological site data are used for the estimation of the HEP. After reexamining the literature, we conclude that the method we used is indeed “spatial block” cross validation, not “bootstrapping”.

- (1) In deriving the HEP model, the archaeological site data are not used directly for resampling, model training and model validation. Instead, the sites are spatially blocked, and the size of the spatial blocks corresponds to the size of the grid cell used for the dynamic modelling of human dispersal, namely, 50 km, as illustrated in Figure 7 (new) in the revised version. It is very important to point out that this spatial-block size corresponds to the scale of “core areas”, a widely used concept used in archaeology, which is a measure of some kind of independence between the archaeological sites (Schmidt and Zimmermann, 2019).
- (2) As repeated stated, we did 1000 runs of model training using randomly selected 80% of the blocks (which we called records) and did the cross-checking using the rest of 20% of the blocks for each run. The goodness of the model is estimated using AUC and BSS averaged over all runs. We recognize that for each training run the 20% of the blocks may still not be fully independent from the 80% of the blocks used for training. Since the data used is the probability of human existence, we expect that there are some spatial correlations in the data. We do not know what the spatial scale is for the correlation but set the block size to be identical to the scale of “core areas” appears to be the most sensible choice to meet the requirement of “independence”. It should be realized that the archaeological data cover a long period of time.

We have tested five additional spatial-block sizes (100 km, 200 km, etc., see Figure 9 new). For these different block sizes, the parameters used to assess the model performance, AUC and BSS, are not that different. As the block size increases, the number of blocks available for model training decreases, causing problems with the modeling training itself. For block size 300 km, the model training does not converge. Another problem with selecting larger blocks is that the estimated HEP must be later spatially interpolated for the dynamic modelling, causing additional uncertainties and numerical cost, making the dynamic simulation of human dispersal over a period of many thousand years numerically costly. Balancing the various factors, we believe the size of the spatial blocks corresponding the grid size of the dynamic modelling and the scale of “core areas” is the most appropriate. We revised the paper and included the discussions as outlined above.

- (3) Our HEP model is a quadratic logistic regression, as illustrated using Figure 8 (new). We assume the spatial blocks in which archeological sites exist as the locations where humans existed with probability 1. For the logistic regression a priori is necessary for conditions under which humans could not exist, namely, with probability of existence 0. We emphasis that, in our view, it is not meaningful to design a totally data driven HEP model. Instead, a priori absence records are needed because humans of different cultures may adapt to different climate/environment conditions. For example, we know Paleolithic humans unlikely lived in the polar region, therefore, HEP there is 0. An exaggerated example is to set HEP = 0 for ocean surface a priori. A HEP model completely data driven can cause problems for the dynamic model of human dispersal, e.g., predict human existence in regions where humans, according to common understanding, could not live.
- (4) The HEP values for the spatial blocks of pseudo absence are estimated using a quadratic logistic regression using climate/environmental predictors. We agree with the referee, there are uncertainties for the estimates for these pseudo-absence spatial blocks. In the revised version, following the suggestion of the referee, we have added a figure (Figure 7c, new) showing the standard deviation of HEP for these blocks, which we estimated from the 1000 training runs.

- (5) The HEP model we then used is not the “final model”, but the “mean model”. We agree there are different choices of the model to use, e.g., using the “best model” from the 1000 training runs, but the “mean model” is more robust. Our approach to quantify the uncertainties of the entire modelling procedure, namely, the HEP model, data, and human dispersal model (HDM) is to do ensemble simulations. For the ensemble generation, different HEP models can be used. However, proper ensemble simulation will require a huge amount of computation. We are now working on this and hope to be able to report the results soon.
- (6) The referee finally suggested to “*use a minimalist climate-potential model in place of the full HEP. For example, if you do have good a priori knowledge of upper and lower temperature thresholds for human habitation, why not just use that as a simple “envelope” model for the climatic constraint on potential human activity and let the OWM and other terrain/environmental submodels do the rest?*” One of our main contributions is to introduce the HEP concept and define different HEPs, climate/environment HEP, assessable HEP and available HEP (see Figure 5). HEP is a key part of our work, not only for static analysis but also for dynamic modelling. Via HEP, we integrate data and model. We know that “minimalist climate-potential models” exist, for example, using NPP (Net Primary Production) as driver for population dynamics, but we know that such models do not work well, because the dynamic model is often insufficiently constrained to perform properly. We know in numerical weather prediction (NWP), highly sophisticated NWP models often drift away from reality quickly. The recent success in NWP is largely due to the data assimilation technique used to constrain the model behavior. In computational science, combining data and model is a highly active research area. What we have done here is an effort to combine data (archaeological and climate/environmental) with a dynamic (human dispersal) model. The key link is HEP.

We very much hope that in this revision and through this reply, we have been able to properly address the comments of the referee, which we truly value.